# Beyond Detection: A Defend-and-Summarize Strategy for Robust and Interpretable Rumor Analysis on Social Media

**Yi-Ting Chang** and **Yun-Zhu Song** and **Yi-Syuan Chen** and **Hong-Han Shuai**

National Yang Ming Chiao Tung University, Taiwan

{joshchang0111.ee10,yzsong.ee07,yschen.ee09,hhshuai}@nycu.edu.tw

## Abstract

As the impact of social media gradually escalates, people are more likely to be exposed to indistinguishable fake news. Therefore, numerous studies have attempted to detect rumors on social media by analyzing the textual content and propagation paths. However, fewer works on rumor detection tasks consider the malicious attacks commonly observed at response level. Moreover, existing detection models have poor interpretability. To address these issues, we propose a novel framework named **D**efend-**A**nd-**S**ummarize (DAS) based on the concept that responses sharing similar opinions should exhibit similar features. Specifically, DAS filters out the attack responses and summarizes the responsive posts of each conversation thread in both extractive and abstractive ways to provide multi-perspective prediction explanations. Furthermore, we enhance our detection architecture with the transformer and Bi-directional Graph Convolutional Networks. Experiments on three public datasets, *i.e.*, RumorEval2019, Twitter15, and Twitter16, demonstrate that our DAS defends against malicious attacks and provides prediction explanations, and the proposed detection model achieves state-of-the-art.[1]

## 1 Introduction

Due to the low cost and easy access to information, social media has become a popular platform for information dissemination. However, it increases the spread of misinformation as well (Vosoughi et al., 2018). The spread of rumors could cause panic and further damage public mental health or lead to severe economic loss (Verma et al., 2022). Therefore, debunking unverified rumors on the Internet has become an indispensable issue (Ahsan et al., 2019). Numerous researchers have been dedicated to detecting rumors automatically. Early works mostly rely on the textual content of each post and the

---

[1]The source code and our datasets are available at https://github.com/joshchang0111/EMNLP2023-RumorDAS.

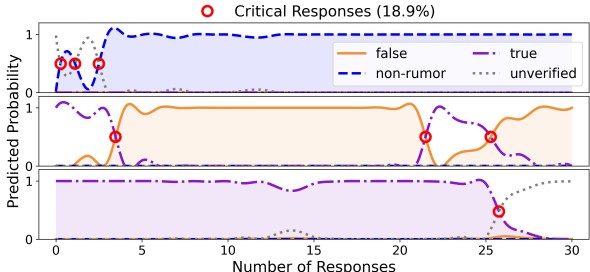

Figure 1: Three examples for the predicted probability of each class with respect to the responses on the Twitter15 dataset. The curves with their face colored represent the ground-truth labels for their source post. Critical responses that result in prediction shifts larger than 0.5 are marked with a red circle.

corresponding responses (Ma et al., 2016; Volkova et al., 2017). In addition, several studies show the importance of considering the propagation path between the responses within the same conversation thread (Ma et al., 2017, 2018; Lu and Li, 2020). To better extract information from the propagation, Graph Convolutional Networks (GCNs) are widely adopted and achieve remarkable performance for the rumor detection problem (Bian et al., 2020; Wei et al., 2021; Sun et al., 2022). For instance, Song et al. (2021) pioneer the integration of transformer and GCN to better detect the rumors.

However, two main challenges remain unaddressed. Firstly, the detectors could be sensitive to *critical responses* toward an event, *i.e.*, responses that significantly impact the detectors. Fig. 1 demonstrates that roughly 18.9% of posts in the Twitter15 dataset contain critical responses. The influence of such responses may be formulated as an attacking manner by adversaries. As prior studies mainly focus on determining the veracity of a given claim by the source post and responsive posts, the potential threat from the attack responses could lead to *vulnerability* in detection models (Le et al., 2020). Hence, some works have developed GAN-

style frameworks to build more robust detectors (Ma et al., 2019, 2021; Song et al., 2021). However, retraining the entire model to defend against attacks could be time-consuming and limited to recognizing only adversarial examples, disregarding various forms of real-world malicious attacks.

On the other hand, recent works mainly leverage neural networks for predictions, making the *interpretability* behind those predictions unattainable due to the black-box property of such models (Ghorbani et al., 2019). To better interpret the detectors' behavior, some works utilize attention mechanism to highlight the important parts of the inputs (Khoo et al., 2020; Lu and Li, 2020), which demonstrates the feasibility of probing the detection models by identifying influential responses in a conversation thread. However, such an approach lacks comprehensive and human-understandable clues, which brings the second challenge of providing organized explanations that cover different viewpoints. We posit that the consideration of multiple perspectives within the discussion threads serves to enhance readers' cognizance of a multitude of viewpoints, thereby discouraging the uncritical acceptance of an excessively confident verdict.

In this paper, we propose a novel framework called **D**efend-**A**nd-**S**ummarize (DAS) to reduce detector vulnerability and provide prediction explanations. The design of DAS follows the idea that responses with similar stances or viewpoints should lie closer in the embedding space. This concept is substantiated by prior studies (Darwish, 2019; Rashed et al., 2021), which showcase that various standpoints of political opinions on Twitter can be well partitioned into distinct clusters based on the embedding representations. This characteristic could enhance summarization with more structured and comprehensive information. As such, DAS includes a response extractor and a response abstractor. The extractor filters and organizes the responses, while the abstractor condenses the information according to the organized responses. To improve robustness, we preemptively mitigate malicious attacks with the response extractor. By exploiting the idea of anomaly detection, we filter the responses by considering the genuine ones as normal data and the attack ones as anomalies. In addition to removing the potentially risky responses, we further organize the remaining responses to find representative ones. We apply clustering to automatically explore the underlying aspects of data to

interpret model predictions from different perspectives. Representative responses are then extracted from the medoid of each cluster. Afterward, the response abstractor aims to produce more comprehensive and human-understandable explanations by summarizing the responses from each cluster. We exploit the pre-trained abstractive summarizers and transfer the models to the rumor detection corpora via self-supervised learning. In particular, the abstractor is finetuned by cluster-summary pairs where the medoid of each response cluster serves as a pseudo summary. Combining extractive and abstractive summaries from DAS provides detectors and users with more reliable and comprehensive information on different viewpoints. Moreover, we introduce a **Bi**-directional **T**ransformer-**G**raph **N**etwork (BiTGN) to improve rumor detection by integrating the robust textual representations of the transformer and the structural information of Bi-directional GCN (BiGCN). The contributions of this paper are summarized as follows:

- We propose a novel framework named DAS that reduces the model vulnerability and provides prediction explanations without additional annotations and retraining of the detection models.

- We explain model predictions with extractive and abstractive summaries by incorporating the concept of clustering into self-supervised learning.

- Experiments on three public datasets show that DAS defends against attacks while producing multi-perspective explanations, and the proposed BiTGN achieves state-of-the-art rumor detection. Human evaluation further demonstrates the interpretability of the generated summaries.

## 2 Related Work

**Model Vulnerability** Adversarial attack has been used to simulate the impact of critical responses (Xu et al., 2021; Mehrabi et al., 2022; Xie et al., 2022). For example, Ma et al. (2019, 2021) develop GAN-style frameworks to generate conflicting utterances that complicate the original conversation threads and force the discriminator to capture more robust text features. Moreover, Le et al. (2020) model such influential responses as a novel attack scenario and propose an adversarial comment generator to mislead the detector. To reduce the model vulnerability to such attacks, Song et al. (2021) perform adversarial training on the detector with

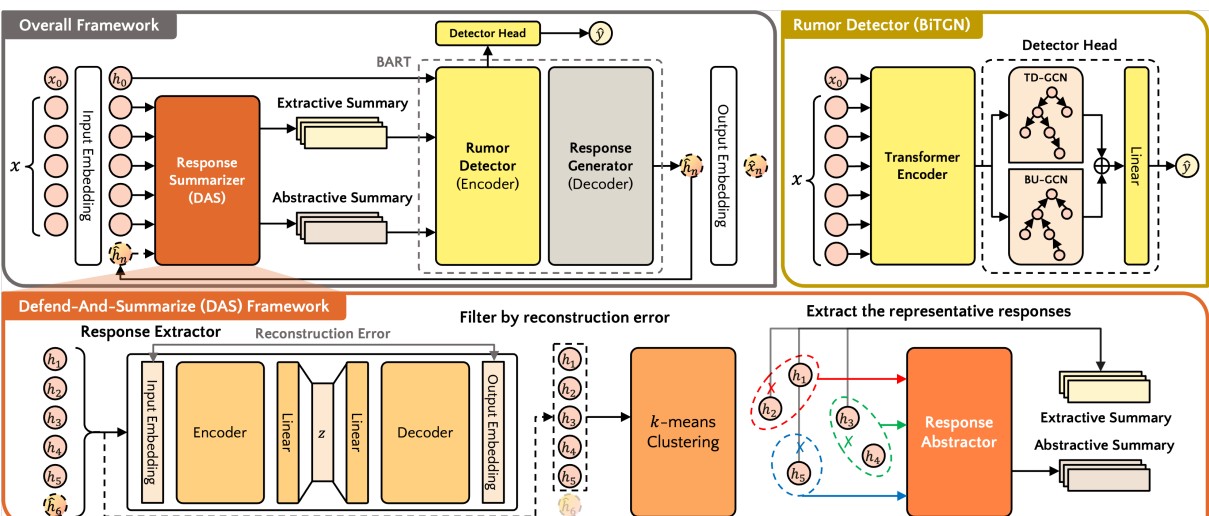

Figure 2: Overview of our proposed framework (upper left). The rumor detector BiTGN (upper right) is trained to predict the veracity of each source post. The response summarizer DAS (lower) preemptively filters out attack responses generated by the response generator. It then organizes the remaining responses into $k$ clusters and produces both extractive and abstractive summaries for each cluster accordingly. During the inference phase, the detector makes predictions based on the source post and the summaries.

adversarial responses. However, this approach requires retraining the entire model. In contrast, our work presents a novel framework that resists response attacks without retraining the model.

**Interpretability** One class of studies typically explains model predictions by analyzing the attention given to different parts of inputs, which is usually accomplished by visualizing the word importance scores (Ribeiro et al., 2016; Vig, 2019) or using heatmap (Samek et al., 2017). For instance, Lu and Li (2020) visualize the attention weights between source tweets and the propagation structures to highlight evidential words and suspicious users in predicting fake news. Similarly, Khoo et al. (2020) provide token-level and post-level explanations by examining the attention weights of transformer layers. Apart from attention-based approaches, Pugoy and Kao (2021) explain recommender system predictions by producing extractive summaries from user and item reviews, which capture crucial sentences for models and provide more comprehensive information than word-level and review-level explanations. Consequently, in this paper, we attempt to provide realistic explanations for rumor detection models by summarizing different opinions in each conversation thread.

**Rumor Detection** Early studies tend to verify the truthfulness of social media posts based on either traditional language processing skills (Badaskar

et al., 2008; Potthast et al., 2018) or hand-crafted features (Yang et al., 2012; Liu et al., 2015; Ma et al., 2015; Wu et al., 2015). In recent years, deep neural networks such as CNN and RNN have been widely adopted to extract the text features automatically (Volkova et al., 2017; Ma et al., 2016). Furthermore, Recursive Neural Networks (RvNN) (Ma et al., 2018) and GCN-based approaches (Bian et al., 2020; Wei et al., 2021; Sun et al., 2022) are proposed to analyze the propagation structures of rumors. In addition, a recent line of studies also leverages Transformers (Khoo et al., 2020; Song et al., 2021; Tian et al., 2022) to capture long-distance interactions between responsive posts.

## 3 Methodology

### 3.1 Problem Formulation

Here, we first define the notations. A conversation thread is denoted by $X = \{x_i\}_{i=0}^n$, where $x_0$ is the source post containing the main event to be verified, and $\{x_i\}_{i=1}^n$ represents the responsive posts of $x_0$. A graph $G = \langle V, E \rangle$ with vertex set $V$ and edge set $E$ is formed by taking each post in $X$ as a node, and the responsive relations between nodes further define the edges. Specifically, two nodes $x_i$ and $x_j$ are connected by an edge $e_{ij} \in E$ if one of them responds to the other one. The ground-truth label is denoted by $y \in Y = \{N, F, T, U\}$ (*i.e.*, Non-Rumor, False Rumor, True Rumor, and Unverified Rumor). In our framework, the rumor detector

aims to predict the veracity of the source post $x_0$ with or without attacks, while the response summarizer aims to extract $k$ representative responses and produce abstractive summaries accordingly.

## 3.2 Rumor Detector

We first introduce the proposed **Bi**-directional **T**ransformer **G**raph **N**etwork (BiTGN) for rumor detection, which integrates the advantages of the transformer network and the Bi-directional Graph Convolutional Networks (BiGCN) as depicted in Fig. 2. Previous studies have shown that Transformer-based models are more robust to out-of-distribution data (Hendrycks et al., 2020) and adversarial attacks (Jin et al., 2020) compared to conventional models such as CNN and RNNs. To obtain a robust textual feature, we adopt a transformer encoder $\theta_{enc}$ with $L_e$ layers to encode all posts in a conversation thread by concatenating them as a sequence. Specifically, the post content is first transformed into vector representations by an embedding layer. Let $\mathbf{h}_i^{(0)} \in \mathbb{R}^{|x_i| \times d}$ denote the embedding of the $i$-th post $x_i$ with dimension $d$. The embedding of a conversation thread $\mathbf{H}^{(0)}$ can be represented as follows,

$$\mathbf{H}^{(0)} = [\mathbf{h}_0^{(0)} \parallel \mathbf{h}_1^{(0)} \parallel ... \parallel \mathbf{h}_n^{(0)}], \quad (1)$$

where $\parallel$ stands for the concatenation. Next, the embeddings are iteratively fed into each encoder layer which consists of Multi-Head Attention (**MHA**) (Vaswani et al., 2017). The hidden representation at the $l$-th transformer layer is denoted by $\mathbf{H}^{(l)} = \mathbf{MHA}(\mathbf{H}^{(l-1)})$. Note that since we concatenate all posts in a thread, different posts can attend to each other and exchange information during the encoding process. After the text encoding process, we obtain the node feature $\mathbf{z}_i$ by taking mean-pooling on all the token representations of the $i$-th post from the last encoder layer. The hidden feature matrix is obtained as follows,

$$\mathbf{Z} = [\mathbf{z}_0 \parallel ... \parallel \mathbf{z}_i \parallel ... \parallel \mathbf{z}_n] \in \mathbb{R}^{n \times d}. \quad (2)$$

To further aggregate the contextual features with the structure of responses, we leverage a GCN-based model $\theta_{gcn}$ to capture the interactions between different posts in two directions, which consists of a Top-Down GCN (TD-GCN) and a Bottom-Up GCN (BU-GCN). Let $\mathbf{A} \in \mathbb{R}^{n \times n}$ denote the adjacency matrix where $\mathbf{A}_{ij} = 1$ if $x_j$ responds to $x_i$. The adjacency matrices for TD-GCN

and BU-GCN are $\mathbf{A}^{TD} = \mathbf{A}$ and $\mathbf{A}^{BU} = \mathbf{A}^{\mathrm{T}}$, respectively. The feature matrix is iteratively updated by each GCN layer in both directions. As such, the aggregated feature $\widehat{\mathbf{Z}}^{TD}$ from the TD-GCN with $L_g$ layers is obtained as follows,

$$\widehat{\mathbf{Z}}^{TD} = \mathrm{MEAN}(\mathbf{Z}_{L_g}^{TD}),$$
$$\mathbf{Z}_l^{TD} = \sigma(\widehat{\mathbf{A}}^{TD} \mathbf{Z}_{l-1}^{TD} \mathbf{W}_{l-1}^{TD}), \quad (3)$$
$$\widehat{\mathbf{A}}^{TD} = \widetilde{\mathbf{D}}^{-\frac{1}{2}} (\mathbf{A}^{TD} + \mathbf{I}) \widetilde{\mathbf{D}}^{-\frac{1}{2}},$$

where $\mathbf{Z}_0^{TD} = \mathbf{Z}$, $\widehat{\mathbf{A}}^{TD}$ is a normalized adjacency matrix with self-connection, $\widetilde{\mathbf{D}}_{ii} = \sum_{j=0} \widetilde{\mathbf{A}}_{ij}^{TD}$ represents the degree of the $i$-th node, and $\mathbf{W}_{l-1}^{TD} \in \mathbb{R}^{d \times d}$ is a learnable matrix. Similarly, the aggregated result for BU-GCN is obtained by substituting $\mathbf{A}^{TD}$ to $\mathbf{A}^{BU}$ in Eq. (3). In the final step, the aggregated features $\widehat{\mathbf{Z}}^{TD}$ and $\widehat{\mathbf{Z}}^{BU}$ are concatenated and passed through a fully connected layer and a softmax function as follows,

$$\widehat{\mathbf{y}} = \mathrm{softmax}([\widehat{\mathbf{Z}}^{TD} \parallel \widehat{\mathbf{Z}}^{BU}] \mathbf{W} + \mathbf{b}), \quad (4)$$

where $\mathbf{W} \in \mathbb{R}^{2d \times |Y|}$ and $\mathbf{b} \in \mathbb{R}^{|Y|}$ are trainable parameters and $\widehat{\mathbf{y}}$ is a vector indicating the predicted probability of each class.

## 3.3 Adversarial Response Generator

To simulate the attack responses from various users in real-world scenarios, we adopt an Adversarial Response Generator (ARG) proposed by Song et al. (2021). Given a conversation thread $\{x_i\}_{i=0}^{n-1}$, ARG generates an adversarial response $x_n^*$ that makes the detector deviate from the ground-truth $\mathbf{y}$ by maximizing the detection loss $\mathcal{L}_{det}$, detailed in Sec. 3.5. Notably, ARG shares the encoder $\theta_{enc}$ with BiTGN and takes the hidden representation of the last encoder layer $\mathbf{H}^{(L_e)}$ as inputs. Moreover, the generated response $x_n^*$ is then attached to the source post $x_0$ to update the adjacency matrix, and its representation $\mathbf{h}_n^*$ is concatenated with the embedding of $\{x_i\}_{i=0}^{n-1}$ to serve as part of the encoder's inputs.[2]

## 3.4 Defend-And-Summarize Framework

**Defensive Response Extractor (DRE)**  To defend against the attacks, we aim to filter out the attack responses simulated by ARG. We hypothesize that if one malicious response can mislead the detector, it must deviate from other normal responses in the embedding space. Therefore, we adopt an autoencoder (AE) to detect the anomalies

---

[2]More details of ARG are provided in the Appendix.

according to the reconstruction error. Concretely, we initialize the encoder $\phi_{ext\text{-}e}$ and decoder $\phi_{ext\text{-}d}$ of the AE with transformer layers and train the model on normal responses. The reconstruction process of a response $x_i$ is as follows,

$$\begin{aligned} \mathbf{z} &= \phi_{ext\text{-}f1}(\phi_{ext\text{-}e}(\mathbf{h}_i)), \\ \widetilde{\mathbf{h}}_i &= \phi_{ext\text{-}d}(\phi_{ext\text{-}f2}(\mathbf{z})), \end{aligned} \quad (5)$$

where $\phi_{ext\text{-}f1}$ and $\phi_{ext\text{-}f2}$ represent fully connected layers. $\mathbf{z} \in \mathbb{R}^{|x_i| \times d_z}$ is the hidden noise with dimension $d_z \ll d$ to compress the features. We apply $\mathcal{L}_2$ loss to calculate the reconstruction error and select the top-$m$ responses with the least loss since an unseen attack response should cause a more significant error. The selection number $m$ is determined by a pre-defined extract ratio $\rho$: $m = \lceil \rho \times n \rceil$. After the filtering process, we take the mean-pooling on the remaining responses $\{\mathbf{h}_i\}_{i=1}^m$ to obtain the response representations $\{\mathbf{r}_i\}_{i=1}^m$. Afterward, $k$-means clustering is performed on these representations to capture the intrinsic perspectives of different responses. The responses are partitioned into $k$ clusters by minimizing the intra-cluster sum of distances from each sample to its nearest centroid. Let $\{C_j\}_{j=1}^k$ denote the set of $k$ clusters. The extractive summary is formed by combining the medoids of all clusters, where a medoid $\mathbf{r}_j^{ext}$ is the response closest to a cluster's centroid $\mathbf{c}_j$:

$$\mathbf{r}_j^{ext} = \underset{\mathbf{r}_i \in C_j}{\arg\min}\{\|\mathbf{r}_i - \mathbf{c}_j\|^2\}. \quad (6)$$

Finally, the embedding of the extraction result is denoted by $\mathbf{H}_{ext} = [\mathbf{h}_1^{ext} \| ... \| \mathbf{h}_j^{ext} \| ... \| \mathbf{h}_k^{ext}]$, where $\mathbf{h}_j^{ext}$ is the response embedding of the medoid $\mathbf{r}_j^{ext}$. Note that some responses may lose their parent node after the extraction. Thus, we assign a new parent node for such response by recursively tracking back until finding a remaining node.

**Self-Supervised Response Abstractor (SSRA)** One main challenge of training the response abstractor is the lack of ground-truth summary labels. Inspired by previous works (Wang and Wan, 2021; Elsahar et al., 2021), we finetune our SSRA $\theta_{abs}$ under self-supervised settings. Previous works often utilize Leave-One-Out (LOO) settings where each response in a conversation thread takes turns to be the pseudo summary. This approach follows the assumption that responses of the same thread focus on the same event, and each of the high-relevance

responses can be approximated as the summary of the whole thread. However, such kind of settings suffers from a great portion of inappropriate responses-summary pairs when the responses cover various aspects. As a result, we create pseudo summaries from the cluster results obtained from DRE. Specifically, each medoid $\mathbf{h}_j^{ext}$ is taken as the pseudo summary, and the remaining responses of cluster $C_j$ are concatenated as the inputs for producing the summary, *i.e.*,

$$\mathbf{h}_j^{abs} = \phi_{abs}([\mathbf{h}_1\|...\|\mathbf{h}_{j-1}\|\mathbf{h}_{j+1}\|...\|\mathbf{h}_{|C_j|}]), \quad (7)$$

where $\mathbf{h}_j^{abs}$ is the summary embedding of cluster $C_j$. $\mathbf{H}_{abs} = [\mathbf{h}_1^{abs} \| ... \| \mathbf{h}_k^{abs}]$ represents the embedding of the abstractive summary. Note that all responses in a cluster are treated as inputs during inference. Besides, each abstractive summary is attached to the source post $x_0$ to maintain the tree structure, where the new adjacency is denoted as $\mathbf{A}'$. Finally, both extractive and abstractive results input along with $x_0$ for rumor detection:

$$\widehat{\mathbf{y}} = \theta_{gcn}(\theta_{enc}([\mathbf{h}_0^{(0)} \| \mathbf{H}_{ext} \| \mathbf{H}_{abs}]), \mathbf{A}'). \quad (8)$$

### 3.5 Training Objectives

We train the rumor detector and adversarial response generator in two stages. The trainable parameters for the detector are the encoder layers $\theta_{enc}$ and the GCN layers $\theta_{gcn}$, while the decoder layers $\theta_{dec}$ are only trained for ARG. In the first stage, the generator is trained with the detector to improve the detection results and its generation quality. The objectives of the generator are the cross entropy for rumor classification $\mathcal{L}_{CE}$ and the cross entropy for text generation $\mathcal{L}_{txt} = -\sum_{i=1}^{|x|} x_i \log \widehat{x}_i$. The objectives of the first stage are calculated as follows,

$$\begin{aligned} \mathcal{L}_{det}(\theta_{enc}, \theta_{gcn}) &= \mathcal{L}_{CE}(\widehat{\mathbf{y}}, \mathbf{y}), \\ \mathcal{L}_{gen}(\theta_{dec}) &= \mathcal{L}_{CE}(\widehat{\mathbf{y}}, \mathbf{y}) + \mathcal{L}_{txt}(\widehat{x}_n, x_n), \quad (9) \\ \mathcal{L}_{1st} &= \mathcal{L}_{det} + \mathcal{L}_{gen}. \end{aligned}$$

In the second stage, we train the generator with a fixed detector. The target of the generator is to produce adversarial response that degrades detector's performance while resembling human writing style. Thus, it is optimized to maximize the cross entropy for rumor detection while minimizing $\mathcal{L}_{txt}$.

$$\begin{aligned} \mathcal{L}_{2nd} &= \mathcal{L}_{gen}(\theta_{dec}) \\ &= -\mathcal{L}_{CE}(\widehat{\mathbf{y}}, \mathbf{y}) + \mathcal{L}_{txt}(\widehat{x}_n, x_n). \end{aligned} \quad (10)$$

| Summarizer | Attack | RE2019 | | | | Twitter15 | | | | Twitter16 | | | |
|---|---|---|---|---|---|---|---|---|---|---|---|---|---|
| | | Acc. | $mF_1$ | ASR↓ | Diff.$_{ASR}$ | Acc. | $mF_1$ | ASR↓ | Diff.$_{ASR}$ | Acc. | $mF_1$ | ASR↓ | Diff.$_{ASR}$ |
| - | - | 64.57 | 64.29 | - | - | 85.64 | 85.50 | - | - | 84.47 | 84.40 | - | - |
| - | ✓ | 33.41 | 19.76 | 62.32 | - | 33.95 | 27.40 | 63.88 | - | 28.12 | 16.80 | 72.19 | - |
| DRE ($\rho$=0.25, $k$=3) | ✓ | 56.51 | 55.82 | 22.35 | -39.97 | 74.98 | 75.62 | 16.39 | -47.49 | 71.76 | 72.37 | 19.58 | -52.61 |
| DRE ($\rho$=0.15, $k$=3) | ✓ | 58.07 | 57.40 | 21.66 | -40.66 | 78.69 | 79.02 | 11.97 | -51.91 | 75.67 | 76.04 | 15.32 | -56.87 |
| DRE ($\rho$=0.05, $k$=3) | ✓ | 59.63 | 58.95 | 19.24 | -43.08 | **80.07** | **80.26** | **10.67** | **-53.21** | **77.02** | **77.25** | **13.74** | **-58.45** |
| DAS ($\rho$=0.25, $k$=3) | ✓ | 57.85 | 57.08 | 21.96 | -40.36 | 72.78 | 73.37 | 18.71 | -45.17 | 70.16 | 70.44 | 21.72 | -50.47 |
| DAS ($\rho$=0.15, $k$=3) | ✓ | **61.22** | **60.33** | **17.93** | **-44.39** | 76.22 | 76.53 | 14.38 | -49.50 | 73.11 | 73.36 | 17.66 | -54.53 |
| DAS ($\rho$=0.05, $k$=3) | ✓ | 60.76 | 59.98 | 17.96 | -44.36 | 77.32 | 77.39 | 13.74 | -50.14 | 74.94 | 75.07 | 15.82 | -56.37 |

Table 1: Overall results of adversarial attack & defense on BiTGN. Diff.$_{ASR}$ represents the difference of ASR with and without defense. Both DRE and DAS can successfully resist a large amount of attacks from ARG.

For the DAS framework, the trainable parameters are the response extractor $\phi_{ext}$ and the response abstractor $\phi_{abs}$. The response extractor is trained to reconstruct the embedding of normal responses by minimizing the $\mathcal{L}_2$ loss between the original embedding $\mathbf{h}_i$ and the reconstructed one $\widetilde{\mathbf{h}}_i$:

$$\mathcal{L}_{ext}(\phi_{ext}) = \mathcal{L}_2(\mathbf{h}_i, \widetilde{\mathbf{h}}_i). \quad (11)$$

The response abstractor is optimized to minimize the cross entropy between the generated summary $s_j^{abs}$ and the pseudo summary $s_j^{ext}$, *i.e.*,

$$\mathcal{L}_{abs}(\phi_{abs}) = \mathcal{L}_{txt}(s_j^{abs}, s_j^{ext}). \quad (12)$$

## 4 Experimental Results

### 4.1 Experimental Setup

**Datasets** We evaluate our model on three real-world public datasets. **Twitter15** and **Twitter16** datasets (Ma et al., 2017) respectively contain 1490 and 818 Twitter posts labeled with Non-Rumor (N), True Rumor (T), False Rumor (F), and Unverified Rumor (U). Moreover, **RumorEval2019 (RE2019)** dataset (Gorrell et al., 2019) was released by the SemEval workshop in 2019, which contains 446 posts from both Twitter and Reddit. It provides three veracity labels, *i.e.*, True Rumor (T), False Rumor (F) and Unverified Rumor (U). The detailed statistics of datasets are listed in the Appendix.

**Evaluation Metrics** For the generation quality of the response abstractor, we calculate the perplexity (PPL) by GPT-2 (Radford et al., 2019) and the factual consistency by FactCC (Kryscinski et al., 2020). For rumor detection, we report the accuracy (Acc.) over all classes, the $F_1$ score of each class and the macro-averaged $F_1$ ($mF_1$).

**Baselines** We compare the performance of rumor detection with several baselines. **RvNN** (Ma et al., 2018) captures the propagation patterns of each

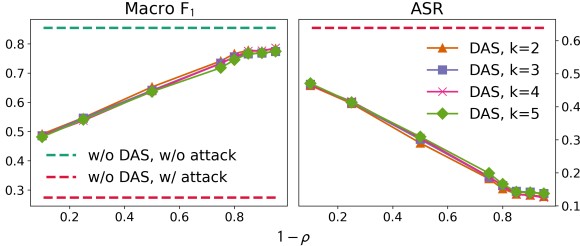

Figure 3: Effect of extract ratio $\rho$ and number of clusters $k$ on Twitter15. The dashed lines represent the detection performance without DAS. The Macro-F1 (left) increases as the extract ratio $\rho$ decreases and the Attack Success Rate (right) behaves in an opposite trend. The number of clusters $k$ does not influence the results significantly, which demonstrates the robustness of DAS.

conversation thread using tree-structured recursive neural networks with GRU units. **BiGCN** (Bian et al., 2020) represents each post with TF-IDF vectors and utilizes a bi-directional GCN to aggregate both propagation and dispersion structures. **EBGCN** (Wei et al., 2021) extends the BiGCN by adaptively updating the weight of each edge formed by the responsive relations with a Bayesian approach. **WETGN** (Song et al., 2021) combines the transformer encoder with a unidirectional GCN and adopts an edge filter. **DUCK**$_{\neg UT}$ (Tian et al., 2022) models each conversation thread as a graph, a stream, and a user tree (UT) separately. Due to the lack of such user information, we eliminate the user tree (¬UT) for a fair comparison. Note that we run these baselines on our version of Twitter15 and Twitter16 datasets since the number of responses is not comparable with previous works.[3]

---

[3]Since Twitter15 and Twitter16 only provide the tweet IDs for responses, we use the version released by Song et al. (2021), where the textual content of each response is manually obtained via Twitter API.

| Summarizer | RE2019 | | Twitter15 | | Twitter16 | |
|---|---|---|---|---|---|---|
| | PPL $\downarrow$ | FactCC | PPL $\downarrow$ | FactCC | PPL $\downarrow$ | FactCC |
| BART-base-SAMSum | **0.63** | 68.77 | **0.87** | 62.40 | **0.55** | 59.48 |
| SSRA-LOO | 4.11 | 71.02 | 4.20 | 71.27 | 7.20 | 73.72 |
| SSRA-$k$-means ($k$=1) | 2.50 | 72.11 | 3.39 | 78.35 | 3.03 | 77.89 |
| SSRA-$k$-means ($k$=2) | 2.59 | 82.29 | 2.41 | **88.13** | 2.55 | 82.97 |
| SSRA-$k$-means ($k$=3) | 2.32 | 83.01 | 2.33 | 87.64 | 2.44 | 82.93 |
| SSRA-$k$-means ($k$=4) | 2.65 | 84.11 | 2.23 | 87.16 | 2.61 | **83.66** |
| SSRA-$k$-means ($k$=5) | 2.70 | **85.74** | 2.14 | 86.88 | 2.39 | 83.03 |

Table 2: Automatic evaluation of generated summaries. The best / second best scores are marked in **bold** / underlined. Our SSRA-$k$-means models generate summaries with better text quality and factual consistency.

## 4.2 Adversarial Attack and Defense

We first discuss how the proposed DAS framework can reduce detector vulnerability. Table 1 demonstrates the results of BiTGN using BART encoder, and under attack by ARG while equipping different response summarizers. Apart from the accuracy and m$F_1$, we also calculate the Attack Success Rate (ASR) of ARG, defined as the ratio of successfully misled predictions among all initially correct predictions. The first and second rows represent the performance before and after the attacks. The results manifest that ARG indeed degrades the detection performance since the attack success rates are greater than 60% on all datasets. Next, we compare the defensive ability of our response summarizers (both DRE and DAS) with different extract ratios $\rho$ given the number of clusters $k$ set to 3. First, the model performance has been recovered significantly by simply equipping DRE during inference, indicating that the extractor can filter out a large portion of the attack responses generated by ARG without retraining. Similarly, DAS can also defend against the attacks while additionally providing prediction explanations with abstractive response summaries. To further analyze the behavior of DAS, Fig. 3 shows the model robustness with different extract ratio $\rho$ and number of clusters $k$ on Twitter15.[4] First of all, the Macro-$F_1$ increases with the extract ratio decreases and saturates when the extract ratio $\rho$ is around 0.2. Besides, even with a high extract ratio, *i.e.*, the left part of both figures, the model could still defend against a certain ratio of attacks while preserving more information from the responses. Secondly, increasing the number of clusters to produce more diverse summaries only slightly affects the defense ability of DAS, which demonstrates the robustness of our model.

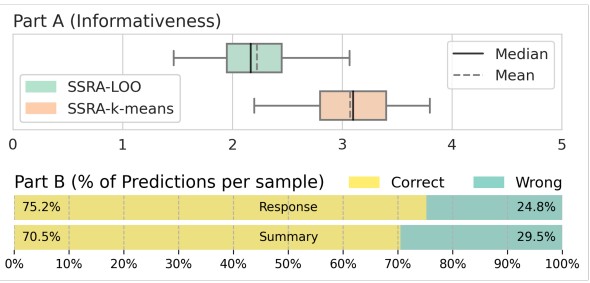

Part A (Informativeness)

Part B (% of Predictions per sample)

Figure 4: Human evaluation of generated summaries. In part A, our SSRA-$k$-means model can generate more informative response summaries compared to SSRA-LOO. In part B, human predictions based on either responses or summaries can achieve comparable accuracy, which demonstrates the interpretability of the summaries.

## 4.3 Interpret Predictions with Summary

**Automatic Evaluation** Here, we show that the generated response summaries can be used to explain model predictions. Since higher text quality can help humans understand models' decisions, we first compare the generation quality of the abstractors trained under different self-supervised settings, including Leave-One-Out (LOO) and $k$-means with varying values of $k$. We initialize all models with BART-base-SAMSum, a summarizer pre-trained on the SAMSum corpus,[5] and take its zero-shot results as a baseline of text quality. Results in Table 2 manifest that self-supervised models demonstrate higher perplexity than BART-base-SAMSum due to the abbreviations and informal expressions in social microblog text, such as hashtags and URLs. Compared to SSRA-LOO, our proposed SSRA-$k$-means achieves better perplexity scores, indicating its ability to generate more fluent and human-understandable summaries. The inclusion of fragmented responses and incomplete sentences in SSRA-LOO training targets contributes to a higher perplexity. Secondly, we validate the factual consistency between the input responses and generated summaries. We observe that SSRA-$k$-means outperforms baselines across different $k$ values, suggesting the necessity of self-supervised learning. Besides, even when $k = 1$, *i.e.*, SSRA-$k$-means only provides one summary as SSRA-LOO does, our model scores higher than SSRA-LOO on all datasets by covering more factual information from the responses. Furthermore, the factuality improves significantly when $k > 1$, demonstrating the effectiveness of providing the abstractor with

---

[4]Due to the space constraint, the results of Twitter16 and RE2019 are provided in the Appendix.

[5]The SAMSum dataset (Gliwa et al., 2019) contains about 16K daily conversations with ground-truth summary labels.

| Source Post (False Rumor): new. leaked phone call between rebel leader & russian intel agent: "cossacks" shot down #MH17. URL URL |
|---|
| Responses ... [5]: @name1 @name2 who leaked this? Do you check sources? Have you verified anything? ... [7]: RT @name1: NEW. Leaked phone call between rebel leader & Russian intel agent: "Cossacks" shot down #MH17. URL ... [15]: @name1 @name3 NEW. Leaked phone call between rebel leader & Russian intel agent: "Cossacks" shot down #MH17 URL ... [19]: @name1 @name4 what is the proof [20]: @name1 Jim, Jim, Jim, shame on you. It was a hoax of sorts and you promoted it Opps there goes our credibility again Too bad @name5 |
| Summary [1]: RT: NEW. Leaked phone call between rebel leader & Russian intel agent: "Cossacks" shot down #MH17 [2]: This is a hoax, it was a hoax of sorts and you promoted it. Shame on you [3]: what is the proof?? |

Table 3: Generation example of SSRA-$k$-means ($k$=3). Key information captured by summaries is highlighted.

| Model | RE2019 | | Twitter15 | | Twitter16 | |
|---|---|---|---|---|---|---|
| | Acc. | m$F_1$ | Acc. | m$F_1$ | Acc. | m$F_1$ |
| RvNN | 52.00 | 51.42 | 73.88 | 73.81 | 75.18 | 75.22 |
| BiGCN | 60.31 | 56.83 | 84.02 | 83.69 | 87.33 | 87.07 |
| EBGCN | 57.43 | 52.91 | 82.79 | 82.48 | 85.43 | 85.16 |
| WETGN | 68.82 | 67.53 | 87.35 | 87.34 | 87.16 | 87.09 |
| DUCK$_{\neg UT}$ | 69.73 | 69.02 | 86.25 | 86.14 | 86.55 | 86.40 |
| BiTGN | **70.84** | **70.05** | **87.77** | **87.73** | **89.10** | **89.02** |
| -BUGCN | 69.27 | 68.45 | 87.76 | 87.72 | 88.02 | 88.00 |
| -TDGCN | 69.50 | 69.03 | 88.39 | 88.33 | 87.78 | 87.79 |
| -GCN | 70.40 | 69.48 | 87.35 | 87.42 | 87.89 | 87.83 |

Table 4: Overall results of rumor detection. The best / second best scores are marked in **bold** / underlined. Our BiTGN outperforms all baselines in the first block.

responses from different perspectives.

**Human Evaluation and Case Study** We recruit 100 human readers and conduct two parts of user study. In part A, we randomly select 10 samples from each of the three datasets, each containing a source post, responses, and two sets of summaries generated by SSRA-LOO[6] and SSRA-$k$-means ($k = 3$). Readers are requested to assess the informativeness of the summaries based on the viewpoint coverage and diversity using a Likert scale from 1 to 5, with 5 representing the most informative. In part B, we aim to evaluate whether humans make a consistent judgment, either after reading responses or response summaries. Thus, we select 20 samples, including 10 true and false rumors, and ask the participants to judge the truthfulness of source posts based on the provided information, *i.e.*, responses or summaries. The upper of Fig. 4 shows that SSRA-$k$-means outperforms SSRA-LOO in terms of informativeness, indicating the effectiveness of utilizing $k$-means clustering to grasp diverse opinions. The results of part B show that participants who solely read the summaries achieve comparable accuracy to those who read the responses on average, with a marginal difference of approximately 5%. Although the responses provide more complete information, these findings suggest that summaries are practical for social media users to judge the post veracity as the summaries effectively capture different viewpoints in a shorter format. In real applications, we could provide both responses and summaries to convey the essential viewpoints by summaries and delve into details in the responses. Moreover, we evaluate the *percentage of ground-truth predictions* $p_{gt}$ for each sample and analyze the correlation

between $p_{gt}$ of responses and summaries. We observe a Pearson correlation of 0.54 with $p$-value 0.014, justifying a high correlation between predictions based on responses and summaries. This demonstrates that the response summaries effectively capture crucial information and can interpret human decisions based on the responses. Table 3 demonstrates a generation example of SSRA-$k$-means ($k = 3$). The source post contains a false claim about a "*Malaysian flight shot down by Cossacks*". Our summaries encompass diverse stances such as "*It was a hoax*" (deny) and "*what is the proof*" (query), providing evidential guidance for models and users to evaluate the veracity of the source post. These summaries also help identify which information models focus on.

### 4.4 Rumor Detection

In this section, we purely analyze the rumor detection results of the proposed BiTGN with RoBERTa encoder in Table 4.[7] Compared with all baselines, our BiTGN achieves the best accuracy and macro-averaged $F_1$ on all datasets. Specifically, transformer-based models (*BiTGN*, *DUCK$_{\neg UT}$*, *WETGN*) outperform models that use TF-IDF vectors (*BiGCN*, *EBGCN*, *RvNN*) as node features, which demonstrates the importance of robust textual representations. Moreover, our model strikes the best among transformer-based baselines, showing that the BiGCN component can better aggregate the conversation information from two directions. Besides, compared to DUCK$_{\neg UT}$ that models the conversation structures with two branches of transformer networks, our model still achieves better results with fewer parameters. We also analyze the influence of GCN by ablating the BiGCN as shown in Table 4. In particular, the model with BiGCN

---

[6]We randomly split the responses into 3 groups and make SSRA-LOO generate one summary for each group.

[7]The detailed statistics are shown in the Appendix.

(*BiTGN*) achieves the best performance on RE2019 and Twitter16 while achieving the second best on Twitter15. We notice that the model without top-down GCN (*-TDGCN*) improves on the classes of non-rumor and true rumor of Twitter15. This may be caused by the diverse structure of these data, as observed by Huang et al. (2020). Although the structure information may be noisy, the model can still benefit from introducing the information of propagation path through GCN layers compared with the model without GCN layers (*-GCN*).

## 5  Conclusion

In this paper, we propose a novel response summarization framework, Defend-And-Summarize (DAS), to enhance the robustness and interpretability of rumor detection models. Our DAS framework is built around two pivotal components: a Defensive Response Extractor (DRE) adept at sifting out malicious responses and extracting significant ones, and a Self-Supervised Response Abstractor (SSRA) capable of producing multi-perspective abstractive response summaries. Moreover, we propose a Bi-directional Transformer Graph Network (BiTGN) to strengthen transformer-based detectors with bi-directional graph aggregation. Experiments on three real-world datasets demonstrate the potent capabilities of DAS to improve the resilience and interpretability of detection models. Besides, BiTGN delivers state-of-the-art rumor detection. The combination of DAS and BiTGN signals promising advancement in the field of rumor detection, providing a robust and interpretable solution primed for future challenges and applications.

## Limitations

Our work focuses on determining the truthfulness of a source post from social media websites by analyzing the structural information of its responses. Since the opinions from various users provide rich information and can influence other readers significantly, we do not consider the settings of fake news detection that solely rely on the news content. Moreover, the DRE component in our framework adopts the widely-used $k$-means algorithm to produce response clusters without considering specific aspects. However, it would be beneficial to create clusters with more fine-grained aspects, such as the stance or sentiment of responses. This would enable humans to gain a more comprehensive understanding of the public's opinion. We'll explore this possibility as a direction for our future work.

## Ethics Statement

We discuss some potential risks that our rumor detection system might raise. As our proposed framework highly relies on the interactions between different users on social media, the content of the users' utterances including mentions to other users will be revealed to the system. However, the system doesn't require any personal information such as user description, user account age, number of followers, number of posts, etc. Due to this reason, the proposed method shall not infringe on individual's privacy. Another risk is that the detector might still give wrong classification results that mislead the users. For this issue, we believe that our method could provide a more simplified but comprehensive summary of the diverse responses under each post, making the users able to observe the opinions on more sides toward an event. This might enhance the ability of the public to rethink and justify the truthfulness of various sources of information.

## Acknowledgements

This work was supported in part by the National Science and Technology Council of Taiwan under Grants NSTC-112-2221-E-A49-059-MY3 and NSTC-111-2221-E-A49-164.

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

# A Adversarial Response Generator

Here, we provide a more detailed formulation of the adversarial response generator. To further simulate the attack responses produced by different users in real-world scenarios, we adopt an adversarial response generator (ARG) proposed by Song et al. (2021), which is trained by adversarial learning under white-box settings. We initialize ARG with a BART model due to its outstanding performance on several text generation tasks. Given a conversation thread $\{x_i\}_{i=0}^{n-1}$, the goal is to generate an adversarial response $x_n^*$ that makes the detector deviate from the ground-truth $\mathbf{y}$ by maximizing the detection loss $\mathcal{L}_{det}$, detailed in Sec. 3.5, *i.e.*,

$$\max_{\mathbf{h}_n^*} \mathcal{L}_{det}(\mathbf{y}^*, \mathbf{y}),$$
$$\mathbf{y}^* = \theta_{gcn}(\theta_{enc}([\mathbf{H}^{(0)} \| \mathbf{h}_n^*]), \mathbf{A}'), \quad (13)$$

where $\mathbf{h}_n^*$ denotes the hidden representation of $x_n^*$ and $\mathbf{A}'$ is the new adjacency that attaches $x_n^*$ to source post $x_0$. To generate a response, we construct the ARG by sharing the encoder $\theta_{enc}$ with BiTGN and feeding the hidden representation of the last encoder layer $\mathbf{H}^{(L_e)}$ to the decoder $\theta_{dec}$ as:

$$\mathbf{h}_n^* = \theta_{dec}(\mathbf{H}^{(L_e)}),$$
$$x_n^* = \text{argmax}(\text{softmax}(\theta_{out}(\mathbf{h}_n^*))). \quad (14)$$

Note that $\theta_{out}$ denotes the output layer, which is tied with the input embedding layer $\theta_{in}$. In this way, $\mathbf{h}_n^*$ can approximate the embedding of a generated response and be concatenated with the embedding of $\{x_i\}_{i=0}^{n-1}$ that serves as part of the encoder's inputs without taking argmax operation. Subsequently, the gradients can be backpropagated from the rumor detection loss to train the ARG. Moreover, the generated response is attached to the source post of the thread and a new edge $e_{0,n}$ between $x_0$ and $\widehat{x}_n$ is thus created.

# B Experimental Setup

## B.1 Datasets

All datasets we used are publicly available. Table 5 displays the statistics of RumorEval2019 (RE2019), Twitter15 and Twitter16 datasets. N, T, F, U represent Non-Rumor, True Rumor, False Rumor and Unverified Rumor respectively. We also calculate the statistics of the number of posts for each claim and report them as *thread length* in the table.

## B.2 Implementation Details

All of our experiments are conducted on a single NVIDIA GeForce RTX 3090 GPU. We conduct 5-fold cross-validation and report the average results for all datasets. The total training time required for each fold on RE2019 and Twitter16 is around one hour, and two hours for Twitter15. The number of trainable parameters is around 300 million. We use the same set of hyperparameters on all datasets. Specifically, the batch size for BiTGN, ARG, and SSRA is 16, while the learning rates are set to $2 \times 10^{-5}$. For DRE, the batch size and learning rate are respectively 256 and $4 \times 10^{-5}$. We finetune BiTGN, ARG and SSRA for 10 epochs, and DRE is trained for 50 epochs. The number of GCN layers $L_g$ in BiTGN is 2, and the encoder and decoder of DRE both consist of 4 transformer layers. The dimension of the hidden noise $\mathbf{z}$ in

| Dataset | RE2019 | Twitter15 | Twitter16 |
|---|---|---|---|
| # of claims | 446 | 1490 | 818 |
| # of N | - | 374 | 205 |
| # of T | 185 | 372 | 207 |
| # of F | 138 | 370 | 205 |
| # of U | 123 | 374 | 201 |
| # of posts | 8529 | 41154 | 18618 |
| Max. thread length | 268 | 304 | 250 |
| Min. thread length | 1 | 2 | 1 |
| Avg. thread length | 19.12 | 28.23 | 22.76 |

Table 5: Dataset Statistics

| Model | RE2019 | | | | | Twitter15 | | | | | | Twitter16 | | | | | |
|---|---|---|---|---|---|---|---|---|---|---|---|---|---|---|---|---|---|
| | Acc. | $mF_1$ | $F_1$-T | $F_1$-F | $F_1$-U | Acc. | $mF_1$ | $F_1$-N | $F_1$-T | $F_1$-F | $F_1$-U | Acc. | $mF_1$ | $F_1$-N | $F_1$-T | $F_1$-F | $F_1$-U |
| RvNN | 52.00 | 51.42 | 53.41 | 54.14 | 46.72 | 73.88 | 73.81 | 70.27 | 81.77 | 72.46 | 70.74 | 75.18 | 75.22 | 67.07 | 85.49 | 74.25 | 74.09 |
| BiGCN | 60.31 | 56.83 | 66.39 | 58.72 | 45.38 | 84.02 | 83.69 | 80.02 | 88.61 | 84.79 | 81.34 | 87.33 | 87.07 | 79.52 | 92.87 | **87.63** | **88.26** |
| EBGCN | 57.43 | 52.91 | 64.73 | 53.67 | 40.33 | 82.79 | 82.48 | 78.01 | 87.97 | 83.87 | 80.05 | 85.43 | 85.16 | 77.01 | 91.36 | 86.61 | 85.65 |
| WETGN | 68.82 | 67.53 | 73.21 | 67.80 | 61.57 | 87.35 | 87.34 | **91.41** | 87.45 | 85.82 | 84.69 | 87.16 | 87.09 | 86.37 | 92.79 | 83.45 | 85.75 |
| DUCK$_{\neg UT}$ | 69.73 | 69.02 | 72.07 | **71.20** | 63.79 | 86.25 | 86.14 | 87.04 | **88.88** | 84.39 | 84.26 | 86.55 | 86.40 | 79.81 | 93.62 | 86.12 | 86.07 |
| BiTGN | **70.84** | **70.05** | **74.30** | 69.70 | **66.15** | **87.77** | **87.73** | 89.82 | 88.63 | **87.34** | **85.15** | **89.10** | **89.02** | **88.78** | **94.52** | 86.13 | 86.66 |
| -BUGCN | 69.27 | 68.45 | 73.72 | 65.94 | 65.67 | 87.76 | 87.72 | 89.93 | 89.44 | 88.04 | 83.48 | 88.02 | 88.00 | 86.20 | 93.47 | 84.66 | 87.66 |
| -TDGCN | 69.50 | 69.03 | 73.45 | 69.35 | 64.31 | 88.39 | 88.33 | 91.31 | 89.93 | 87.06 | 85.02 | 87.78 | 87.79 | 86.89 | 95.59 | 81.77 | 86.91 |
| -GCN | 70.40 | 69.48 | 74.54 | 68.51 | 65.41 | 87.35 | 87.42 | 90.96 | 86.68 | 87.15 | 84.90 | 87.89 | 87.83 | 87.10 | 94.18 | 84.14 | 85.89 |

Table 6: Detailed results of rumor detection. The $F_1$ score of each class is reported. The best / second best scores of the first two blocks are marked in **bold** / underlined. Our BiTGN outperforms all baselines listed in the first block.

| Model | RE2019 | | Twitter15 | | Twitter16 | |
|---|---|---|---|---|---|---|
| | $p$ (Acc.) | $p$ ($mF_1$) | $p$ (Acc.) | $p$ ($mF_1$) | $p$ (Acc.) | $p$ ($mF_1$) |
| RvNN | $1.29 \times 10^{-4}$ | $1.30 \times 10^{-4}$ | $6.79 \times 10^{-5}$ | $7.60 \times 10^{-5}$ | $1.14 \times 10^{-4}$ | $1.15 \times 10^{-4}$ |
| BiGCN | $9.29 \times 10^{-3}$ | $5.52 \times 10^{-3}$ | $3.14 \times 10^{-3}$ | $1.42 \times 10^{-3}$ | $1.31 \times 10^{-1}$ | $1.30 \times 10^{-1}$ |
| EBGCN | $4.14 \times 10^{-3}$ | $4.25 \times 10^{-3}$ | $1.29 \times 10^{-3}$ | $1.38 \times 10^{-3}$ | $2.52 \times 10^{-2}$ | $2.39 \times 10^{-2}$ |
| WETGN | $2.25 \times 10^{-1}$ | $1.69 \times 10^{-1}$ | $3.32 \times 10^{-1}$ | $3.37 \times 10^{-1}$ | $1.27 \times 10^{-1}$ | $1.37 \times 10^{-1}$ |
| DUCK$_{\neg UT}$ | $3.57 \times 10^{-1}$ | $3.57 \times 10^{-1}$ | $1.13 \times 10^{-1}$ | $1.00 \times 10^{-1}$ | $1.09 \times 10^{-1}$ | $1.12 \times 10^{-1}$ |

Table 7: Paired student t-test between our BiTGN and other baselines for rumor detection. The $p$-value of both accuracy (Acc.) and macro-averaged $F_1$ ($mF_1$) are presented.

DRE is set to 100. Moreover, we implement our framework with Hugging Face Transformers and PyTorch. For the results of rumor detection in section 4.4, the transformer encoder of BiTGN is initialized with RoBERTa-base[8]. For the results of adversarial attack and defense in section 4.2, the ARG shares the same encoder with BiTGN, and the overall encoder-decoder framework is initialized with BART-base[9]. The SSRA is initialized with BART-base-SAMSum[10]. We also follow (Song et al., 2021) to perform tree decomposition on the original datasets where each conversation thread is decomposed into several subtrees by adding each response one by one in chronological order. In this way, we not only increase the amount of training data for BiTGN but also create a pseudo-ground-truth response for ARG from the last response of each subtree. For the baseline models of rumor detection, we use the official implementation for RvNN[11], BiGCN[12], EBGCN[13] and DUCK$_{\neg UT}$[14]. For WETGN[15], we implement the model architecture by ourselves due to the similar design. For the evaluation of factual consistency, we use the

official implementation of FactCC[16].

## C  Additional Results of Rumor Detection

### C.1  Models with Different Backbones

We provide the results of WETGN and BiTGN using different transformer encoders, including RoBERTa with 12 self-attention layers and BART encoder with 6 self-attention layers. The results are shown in Table 8. It is obvious that both WETGN and BiTGN perform better with RoBERTa encoder, which is expected since RoBERTa contains more layers than BART encoder. Moreover, RoBERTa is pre-trained on several text classification tasks, while BART is more effective on text generation tasks. We also report the rumor detection results of BiTGN after each adversarial training stage. For the first stage (BiTGN†), the inputs of the detector contain the response generated from ARG, and the second stage (BiTGN*) contains only the responses from original data. We can observe that the model performs better in the first stage, which indicates that the generated response in the first stage can help the detector improve its accuracy.

### C.2  Detailed Detection Results

Table 6 provides the detailed results of rumor detection, including the $F_1$ score for each class. The results demonstrate that BiTGN outperforms all baselines listed in the first block, demonstrating

[8] https://huggingface.co/roberta-base

[9] https://huggingface.co/facebook/bart-base

[10] https://huggingface.co/lidiya/bart-base-samsum

[11] https://github.com/majingCUHK/Rumor_RvNN

[12] https://github.com/TianBian95/BiGCN

[13] https://github.com/weilingwei96/EBGCN

[14] https://github.com/ltian678/DUCK-code

[15] https://github.com/yunzhusong/AARD

[16] https://github.com/salesforce/factcc

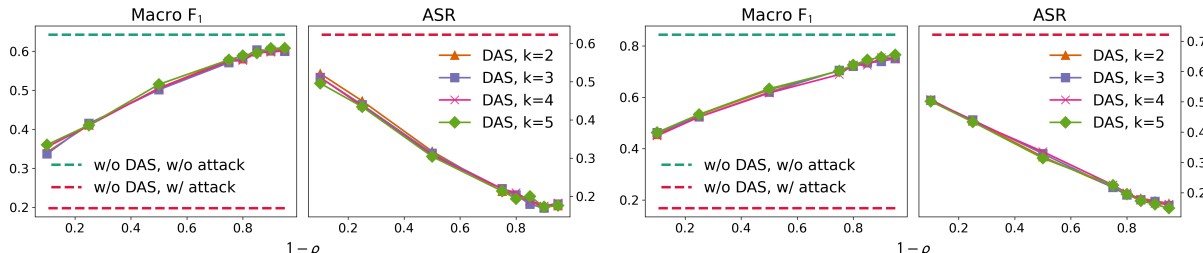

Figure 5: Effect of extract ratio $\rho$ and number of clusters $k$ on RE2019 (left) and Twitter16 (right). The dashed lines represent the detection performance without DAS. The Macro-$F_1$ increases as the extract ratio $\rho$ decreases on both datasets, and the Attack Success Rate (ASR) behaves in an opposite trend. Moreover, the number of clusters $k$ do not influence the results significantly, demonstrating the robustness of DAS.

| Model | Backbone | RE2019 | | Twitter15 | | Twitter16 | |
|---|---|---|---|---|---|---|---|
| | | Acc. | m$F_1$ | Acc. | m$F_1$ | Acc. | m$F_1$ |
| WETGN | BART | 67.03 | 65.59 | 86.53 | 86.40 | 85.82 | 85.71 |
| WETGN | RoBERTa | 68.82 | 67.53 | 87.35 | 87.34 | 87.16 | 87.09 |
| BiTGN$^\dagger$ | BART | 66.14 | 65.69 | 86.94 | 86.79 | 86.06 | 85.95 |
| BiTGN* | BART | 64.79 | 64.46 | 86.12 | 85.98 | 85.81 | 85.76 |
| BiTGN | RoBERTa | **70.84** | **70.05** | **87.77** | **87.73** | **89.10** | **89.02** |

Table 8: Performance of BiTGN and WETGN with BART and RoBERTa as transformer encoder. RoBERTa encoder improves the performance on all models.

its power contributed by the robust textual representations from the transformer network and the effective graph aggregation from the BiGCN component. Moreover, as discussed in section 4.4, we see that BiTGN has a lower $F_1$ score on the classes of non-rumor and true rumor of Twitter15, which may be caused by the diverse structural information of this dataset as observed by Huang et al. (2020). To validate the significance of the detection results, we perform the paired student t-test between the proposed BiTGN and each of the baseline models. The corresponding $p$-values for both accuracy and macro-averaged $F_1$ scores are presented in Table 7. Notably, our BiTGN not only achieves the best average accuracy and $F_1$ scores as discussed previously but also significantly outperforms models utilizing TF-IDF vectors as node features (BiGCN, EBGCN, RvNN). We notice that the significance levels for the comparisons with WETGN and DUCK$_{\neg UT}$ are relatively lower. This could be attributed to their shared utilization of a transformer backbone and the notable variability in folds. Nonetheless, in comparison to WETGN, which incorporates a top-down GCN with weighted edges, our BiTGN effectively benefits from the bi-directional GCN component, even in the absence of weighted edges. When compared against DUCK$_{\neg UT}$, which employs two distinct branches

of transformers to independently model each conversation thread as both a stream and a graph, our model achieves a more favorable average performance by utilizing only a single transformer branch, thereby resulting in fewer parameters.

## D  Adversarial Attack and Defense on Other Datasets

We investigate the robustness of DAS under different extract ratio $\rho$ and number of clusters $k$ on RE2019 and Twitter16 datasets, as illustrated in Fig. 5. The dashed lines represent the detector's performance without equipping the DAS framework. Specifically, the red line and green line stands for the performance of the model with and without being attacked respectively. The results of all datasets are intuitive and similar, where the $F_1$ score increases as the extract ratio $\rho$ decreases, and the Attack Success Rate (ASR) behaves in an opposite trend. Moreover, increasing the number of clusters $k$ to provide responses from more perspectives do not affect the model performance drastically.

## E  Additional Results of DAS Framework

### E.1  Rumor Detection with DAS

To observe how the proposed response summarization framework affects the performance of rumor detection, we provide the detection results of BiTGN equipped with the variants of DAS without being attacked in Table 9. DRE$^\dagger$ denotes DRE with the autoencoder (AE) only. In this experiment, BiTGN is initialized with BART-base, and the number of clusters $k$ is set to 3, *i.e.*, DRE extracts 3 responses, and DAS further produces 3 abstractive summaries based on the 3 clusters. Firstly, we observe that both DRE$^\dagger$ and DRE can approximate the model's performance. Notably, DRE$^\dagger$

| Summarizer | RE2019 | | | | | Twitter15 | | | | | | Twitter16 | | | | | |
|---|---|---|---|---|---|---|---|---|---|---|---|---|---|---|---|---|---|
| | Acc. | $mF_1$ | $F_1$-T | $F_1$-F | $F_1$-U | Acc. | $mF_1$ | $F_1$-N | $F_1$-T | $F_1$-F | $F_1$-U | Acc. | $mF_1$ | $F_1$-N | $F_1$-T | $F_1$-F | $F_1$-U |
| - | 64.57 | 64.29 | 65.25 | 64.01 | 63.63 | 85.64 | 85.50 | 88.52 | 87.75 | 85.35 | 80.37 | 84.47 | 84.40 | 85.62 | 92.16 | 80.78 | 79.06 |
| DRE$^\dagger$ ($\rho$=0.05) | 60.08 | 59.88 | 62.22 | 60.50 | 56.93 | 83.78 | 83.68 | 85.51 | 86.49 | 83.43 | 79.31 | 82.39 | 82.31 | 81.88 | 91.80 | 76.81 | 78.73 |
| DRE$^\dagger$ ($\rho$=0.10) | 60.67 | 60.10 | 64.32 | 58.74 | 57.24 | 84.40 | 84.31 | 87.59 | 86.71 | 83.42 | 79.52 | 82.52 | 82.45 | 82.35 | 92.03 | 76.79 | 78.61 |
| DRE$^\dagger$ ($\rho$=0.15) | 60.76 | 60.41 | 62.54 | 60.77 | 57.92 | 84.54 | 84.43 | 87.49 | 86.85 | 84.12 | 79.27 | 83.13 | 83.07 | 83.63 | 91.88 | 77.92 | 78.84 |
| DRE$^\dagger$ ($\rho$=0.20) | 61.43 | 61.10 | 63.33 | 62.10 | 57.86 | 84.67 | 84.54 | 87.40 | 86.48 | 84.23 | 80.08 | 82.89 | 82.83 | 84.02 | 90.75 | 77.54 | 79.03 |
| DRE$^\dagger$ ($\rho$=0.25) | 61.88 | 61.41 | 63.53 | 64.25 | 56.45 | 84.95 | 84.86 | 87.59 | 86.62 | 84.64 | 80.70 | 83.13 | 83.08 | 84.02 | 90.64 | 78.26 | 79.31 |
| DRE$^\dagger$ ($\rho$=0.50) | 63.00 | 62.40 | 64.54 | 64.31 | 58.36 | 86.12 | 85.96 | 89.08 | 88.42 | 85.62 | 80.73 | 83.38 | 83.37 | 90.40 | 83.96 | 78.14 | 81.00 |
| DRE$^\dagger$ ($\rho$=0.75) | 64.35 | 63.69 | 66.14 | 64.94 | 59.98 | 85.70 | 85.56 | 88.64 | 88.09 | 85.44 | 80.07 | 84.96 | 84.94 | 85.60 | 92.45 | 80.11 | 81.60 |
| DRE$^\dagger$ ($\rho$=0.90) | 64.12 | 63.49 | 65.21 | 64.16 | 61.10 | 85.91 | 85.77 | 88.90 | 87.90 | 85.58 | 80.68 | 84.22 | 84.12 | 83.78 | 91.98 | 79.75 | 80.99 |
| DRE ($\rho$=0.05) | 60.08 | 59.88 | 62.22 | 60.50 | 56.93 | 83.78 | 83.68 | 85.51 | 86.49 | 83.43 | 79.31 | 82.39 | 82.31 | 81.88 | 91.80 | 76.81 | 78.73 |
| DRE ($\rho$=0.10) | 60.53 | 60.22 | 62.93 | 61.15 | 56.58 | 84.19 | 84.11 | 87.16 | 86.71 | 83.10 | 79.45 | 82.76 | 82.69 | 82.81 | 92.03 | 77.11 | 78.80 |
| DRE ($\rho$=0.15) | 60.31 | 59.97 | 62.15 | 60.03 | 57.72 | 84.54 | 84.45 | 87.66 | 86.85 | 83.65 | 79.63 | 83.01 | 82.95 | 83.36 | 91.88 | 77.71 | 78.84 |
| DRE ($\rho$=0.20) | 61.43 | 61.11 | 63.35 | 62.31 | 57.66 | 84.95 | 84.86 | 87.59 | 86.97 | 84.57 | 80.33 | 82.88 | 82.84 | 83.41 | 90.64 | 77.52 | 79.80 |
| DRE ($\rho$=0.25) | 61.88 | 61.50 | 63.77 | 64.42 | 56.30 | 85.15 | 85.07 | 87.49 | 87.01 | 85.19 | 80.59 | 83.13 | 83.08 | 83.98 | 91.11 | 78.06 | 79.17 |
| DRE ($\rho$=0.50) | 62.10 | 61.70 | 64.02 | 61.09 | 59.97 | 84.95 | 84.88 | 87.65 | 87.04 | 84.47 | 80.37 | 82.89 | 82.84 | 82.50 | 90.21 | 77.48 | 81.15 |
| DRE ($\rho$=0.75) | 61.88 | 61.65 | 62.53 | 63.52 | 58.91 | 84.67 | 84.57 | 87.83 | 87.03 | 84.39 | 79.03 | 84.22 | 84.23 | 85.51 | 91.98 | 78.49 | 80.96 |
| DRE ($\rho$=0.90) | 61.88 | 61.46 | 63.75 | 62.28 | 58.35 | 84.95 | 84.83 | 88.00 | 87.10 | 84.61 | 79.60 | 83.62 | 83.61 | 83.65 | 91.18 | 78.71 | 80.92 |
| DRE ($\rho$=1.00) | 59.64 | 59.35 | 60.98 | 60.50 | 56.58 | 84.54 | 84.45 | 87.32 | 86.72 | 84.09 | 79.66 | 83.12 | 83.11 | 83.34 | 91.56 | 78.26 | 79.28 |
| DAS ($\rho$=0.05) | 61.66 | 60.99 | 64.81 | 61.22 | 56.93 | 81.31 | 81.13 | 82.20 | 84.88 | 80.50 | 76.97 | 80.93 | 80.73 | 78.42 | 90.60 | 76.22 | 77.68 |
| DAS ($\rho$=0.10) | 62.56 | 61.88 | 65.87 | 61.08 | 58.70 | 81.72 | 81.53 | 82.20 | 85.20 | 81.05 | 77.65 | 81.42 | 81.25 | 79.19 | 91.05 | 77.47 | 77.30 |
| DAS ($\rho$=0.15) | 62.11 | 61.65 | 64.10 | 61.51 | 59.34 | 82.13 | 81.97 | 84.02 | 84.80 | 80.90 | 78.15 | 81.30 | 81.11 | 79.63 | 89.44 | 77.12 | 78.24 |
| DAS ($\rho$=0.20) | 63.00 | 62.29 | 65.35 | 62.98 | 58.54 | 82.75 | 82.60 | 84.39 | 84.82 | 81.80 | 79.37 | 81.05 | 80.99 | 80.38 | 88.32 | 76.29 | 78.95 |
| DAS ($\rho$=0.25) | 63.90 | 63.09 | 67.15 | 64.14 | 57.97 | 82.41 | 82.26 | 83.54 | 85.29 | 81.60 | 78.61 | 82.28 | 82.19 | 80.97 | 89.54 | 78.72 | 79.53 |
| DAS ($\rho$=0.50) | 63.45 | 62.54 | 66.85 | 63.28 | 57.50 | 83.09 | 82.98 | 84.37 | 86.54 | 81.83 | 79.18 | 82.16 | 82.10 | 79.71 | 89.51 | 79.98 | 79.22 |
| DAS ($\rho$=0.75) | 63.68 | 62.94 | 67.14 | 61.87 | 59.80 | 83.09 | 82.98 | 86.02 | 86.54 | 81.12 | 77.79 | 82.51 | 82.43 | 81.25 | 90.97 | 77.34 | 80.15 |
| DAS ($\rho$=0.90) | 63.67 | 62.73 | 66.67 | 61.55 | 59.96 | 83.09 | 82.94 | 84.91 | 86.89 | 81.68 | 78.30 | 83.25 | 83.19 | 81.15 | 90.41 | 79.90 | 81.28 |
| DAS ($\rho$=1.00) | 64.34 | 63.44 | 67.31 | 63.40 | 59.60 | 83.16 | 83.06 | 84.84 | 87.59 | 81.31 | 78.53 | 82.64 | 82.54 | 82.07 | 91.31 | 77.42 | 79.36 |

Table 9: Rumor detection results of BiTGN using different summarizers without being attacked. The results of DRE$^\dagger$, DRE, and DAS are demonstrated in the upper, middle, and lower sections, respectively. DRE$^\dagger$ represents DRE with the autoencoder (AE) only. The number of clusters $k$ is set to 3. Both DRE$^\dagger$ and DRE can approximate the detection results of BiTGN without summarizers, and DAS only slightly degrades the performance.

even scores higher on the Twitter15 and Twitter16 datasets, potentially due to the filtering process, which identifies and removes noisy responses that could degrade the model's performance. Moreover, DRE achieves comparable results even if only 3 responses are selected, indicating that the extractor can effectively capture the representative responses from each conversation thread. As such, the extracted responses can be used to interpret the model's behavior. Next, for all summarizers, the performance rises as the extract ratio $\rho$ increases and saturates at different ratios on different datasets, which suggests that the representative responses could be potentially excluded if we filter out too many of them. Lastly, we also find that DAS slightly degrades the detection performances, possibly due to the distribution shift of abstractive summaries as the abstractor is tuned under self-supervised settings without fine-grained ground-truth labels. Furthermore, the unsatisfying quantity of data also indicates that there is still room for improvement. We consider enhancing the text quality under self-supervised settings as a potential avenue for future research.

## E.2 Human Evaluation

In this section, we further provide the settings and additional results of human evaluation. We recruit 100 human readers for both parts of the human evaluation. We additionally insert a trick question in part A to validate the answer quality of each participant, and each participant will be paid $5 if he / she passes the trick question. To verify the consistency of the collected results, we calculate the Fleiss' Kappa to evaluate the inter-rater reliability of the human evaluation. For part A, we calculate the Fleiss' Kappa to assess the readers' agreement toward rating our model more favorably than the baseline, and the score is 0.3321. In part B, for readers that make predictions based on the responses and summary, we obtain the Fleiss' Kappa scores of 0.5341 and 0.4225, respectively. Following the criteria outlined by Landis and Koch (1977), these values indicate a fair and moderate agreement among participants in part A and B. It's important to note that this level of agreement has been established as reliable in prior research studies (Cao and Wang, 2021; Chen et al., 2021). In part A, we select SSRA-LOO as the baseline and compare its informativeness with SSRA-$k$-means ($k$=3). Since

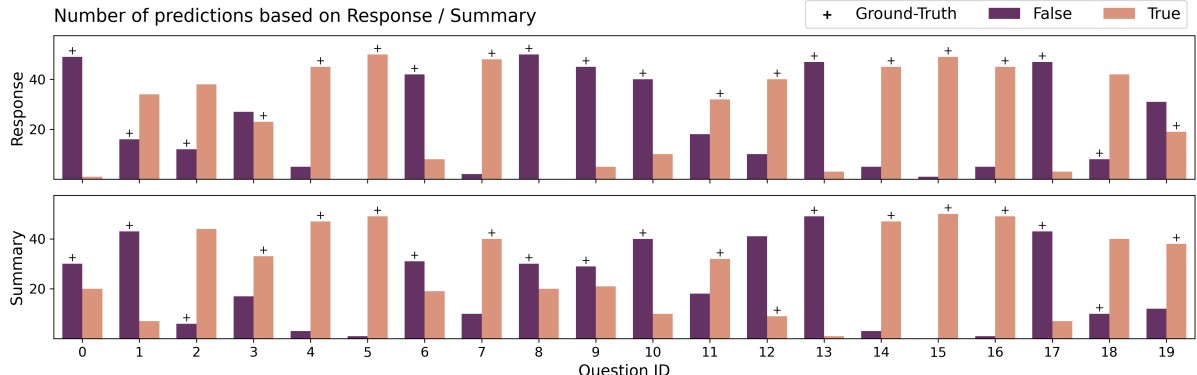

Figure 6: Visualization of the number of predictions based on the responses / summary for each sample of human evaluation part B. Ground-truth label for each sample is marked with "+". In most cases, predictions based on the responses and the summaries are highly correlated, demonstrating the interpretability of the summaries.

SSRA-LOO generates only one summary by default, we randomly divide each conversation thread into 3 groups and make SSRA-LOO generate one summary for each group for a fair comparison. We ask the participants to rate each set of summaries based on the following scoring strategy:

- **Score 1**: None of the three summaries accurately capture the information from the responses, and the summaries may repeat the same information or contain unrelated information.

- **Score 2**: Only one summary captures the information from the responses, but the other ones may be incomplete, inaccurate, or repetitive.

- **Score 3**: Most of the summaries accurately capture the information from the responses, but some important perspectives may be missing, or some information may be repeated unnecessarily.

- **Score 4**: All three summaries accurately capture the information from the responses but may not fully cover all important perspectives or provide a nuanced understanding of the issue.

- **Score 5**: All three summaries accurately capture the information from the responses, and the summaries provide a comprehensive and nuanced understanding of the issue. The summaries cover diverse perspectives and avoid repetition.

In part B, we select 20 samples from RE2019 and Twitter15 datasets, with 5 true rumors and false rumors from each of them. The participants are asked to determine whether a source post is true or false based on all its responses or a set of summaries. A key idea is to observe whether the responses or

summaries deny the rumor, but the readers are not required to accept all the utterances and can make their decisions based on their intuition after reading the provided information. We visualize the predictions based on the responses and summary in Fig. 6. Notably, a higher number of individuals accurately predict the veracity ($p_{gt} > 0.5$) of 15 and 17 samples based on the responses and summary, respectively. Despite the slightly lower average accuracy of the summary, as shown in Fig. 4, the results still indicate its effectiveness in providing social media users with essential information from the responses. Additionally, we observe a high correlation between the predictions obtained from the responses and summaries in most cases. This emphasizes the interpretability of the summaries, as we can identify the crucial information that the detection models focus on when making predictions.

### E.3 Generation Examples

We demonstrate more examples by DAS in Table 10, and 11. QID corresponds to the Question ID of part B human evaluation, as shown in Fig. 6. We provide the source post, responses, and both extractive and abstractive summaries for each example. The responses are arranged in different clusters (*i.e.*, Cluster 1, Cluster 2, and Cluster 3) and chronological order (*i.e.*, [1], [2], ...). Specifically, we highlight the crucial information of the responses captured by the summaries with different colors for each cluster. The results show that both extractive and abstractive summaries can capture essential viewpoints from different perspectives. Moreover, the responses in the same clusters contain similar information, indicating the clustering mechanism

in DAS can indeed identify the underlying aspects in the diverse responses of a conversation thread. The summaries can not only explain predictions from rumor detection models but also benefit social media readers and make them quickly understand the public's opinions toward specific events.

| QID | Content |
|---|---|
| | **Source Post (False Rumor)**: a claim that #obama used the #shutdown to scuttle the amber alert system reveals an ignorance about amber alerts. URL |
| 9 | **Responses (Cluster 1)** [2]: "@name1 : Claim #Obama used #shutdown to scuttle AmberAlerts reveals ignorance abt AmberAlerts URL" @name2 [5]: RINO!!! RT @name1 A claim that #Obama used the #shutdown to scuttle the Amber Alert system reveals an ignorance about Amber Alerts. [6]: @name1 Thebn what was he doing when he shut it down? [9]: @name1 @name3 you retweeted ignorance today. "Amber alert ..." False [10]: @name1 No it doesn't It shows how ignorant ppl are for believing stupid S # % t #SHUTDOWN **(Cluster 2)** [1]: @name1 Just got asked to "inform" someone about this - since he/she can't be bothered to inform self. URL **(Cluster 3)** [3]: @name1 @name4 Perhaps, but a smart administration wouldn't have put that notice on the site. I have no sympathy. [4]: @name1 @name4 That's not the claim. The claim is that Obama is trying to scare people by appearing to shut down sites. [7]: @name1 @name5 I'm really sick and tired of seeing more of this fake news finding traction; how stupid are these readers? [8]: @name1 @name6 Why care at all?? Maybe the little scamps wandered off to look for some food since their SNAP was cut. Just a guess. [11]: @name7 I'm willing to bet there's a high probability that the people who believe this might also believe Obama is Kenyan [12]: @name1 @name8 These lies about the #GOPShutdown are not isolated. They're part of RNC strategy of distraction URL |
| | **Extractive Summary** [1]: @name1 No it doesn't It shows how ignorant ppl are for believing stupid S # % t #SHUTDOWN [2]: @name1 Just got asked to "inform" someone about this - since he/she can't be bothered to inform self. URL [3]: @name1 @name4 Perhaps , but a smart administration wouldn't have put that notice on the site. I have no sympathy. |
| | **Abstractive Summary** [1]: Claim #Obama used #shutdown to scuttle Amber Alerts reveals ignorance abt AmberAlerts [2]: I don't know how to inform someone about this - since he/she can't be bothered to inform self. [3]: That's not the claim. The claim is that Obama is trying to scare people by appearing to shut down sites. |
| | **Source Post (True Rumor)**: krispy kreme hull is advertising kkk wednesday. i don't know. i do not know. URL |
| 11 | **Responses (Cluster 1)** [1]: @name1 Via Burlingame. URL [4]: @name1 @name2 Is that a real ad? [6]: @name1 sprinkled with WHITE POWDER? [9]: Anybody could make that... @name1 @name3 [11]: @name1 Ack! [12]: @name4 Hull, we need to talk... [16]: @name1 [17]: @name1 seriously, how could someone had been so stupid. [19]: @name1 I wonder if they use... URL **(Cluster 2)** [2]: "@name1: Krispy Kreme hull is advertising KKK Wednesday. I don't know. I do not know. URL" #Fail [3]: @name1 @name2 Its a company born in the dirty south, sooo? [5]: Wow! RT "@name1: Krispy Kreme hull is advertising KKK Wednesday. I don't know. I do not know. URL" [14]: @name5 @name1 Which is really good. Unfortunately, the mockery will continue until morale improves. [18]: @name1 @name6 Krispy Kreme is headquartered in the American South, (Winston Salem, North Carolina) sooo... [20]: @name1 @name7 The Kicker is that this was advertised to children as Krispy Kreme Klub, teaching racism and poor spelling all in one. **(Cluster 3)** [7]: @name1 Hi, we know we got it wrong & wholeheartedly apologise. We're taking steps to make sure it doesn't happen again [8]: @name1 They're dull in Hull and the Isle of Mull is seething with discontent. [10]: @name1 GG Hull, glad to know the 2017 city of culture is trying to be inclusive of all groups [13]: @name1 Like the article said. It was a poor choice of the play on the word Club (*spelled Klub) Their intentions were good though. [15]: @name5 @name1 you're going to take step to make sure you don't create any more sales events with unbelievably racist names? Okay. |
| | **Extractive Summary** [1]: @name1 Via Burlingame. URL [2]: Wow! RT "@name1: Krispy Kreme hull is advertising KKK Wednesday. I don't know. I do not know. URL" [3]: @name1 GG Hull, glad to know the 2017 city of culture is trying to be inclusive of all groups |
| | **Abstractive Summary** [1]: That's so stupid!!! [2]: Krispy Kreme hull is advertising KKK Wednesday. I don't know. I do not know. [3]: I'm sorry to hear this, but it was a poor choice of the play on the word Club (*spelled Klub). |

Table 10: Generated examples of DAS ($k$=3). QID corresponds to the Question ID in Fig. 6. The responses are arranged in different clusters and chronological order. Key information captured by summaries is highlighted with different colors for each cluster. The responses within the same cluster deliver similar information, and the produced summaries can effectively capture essential information from the responses.

| QID | Content |
|-----|---------|
| 13 | **Source Post (False Rumor):** new. leaked phone call between rebel leader & russian intel agent: " cossacks šhot down #mh17. URL URL |
| | **Responses (Cluster 1) [5]:** @name1 @name2 who leaked this? Do you check sources? Have you verified anything? **[6]:** "Fuck them. They should not fly, we are at war here." RT @name1: NEW. Leaked phone call between rebel leader & Russian intel agent #MH17 **[7]:** RT @name1: NEW. Leaked phone call between rebel leader & Russian intel agent: "Cossacks" shot down #MH17. URL ... **[8]:** 1/2 RT @name1 Leaked phone call/rebel leader & Russian intel agent: "Cossacks" shot down #MH17. URL URL **[12]:** @name1 It not seems as a reliable source. Now all are fabricating they own versions. **[14]:** @name1 @name3 careful what you believe; way too much propaganda and lies out there **[15]:** @name1 @name4 NEW. Leaked phone call between rebel leader & Russian intel agent: "Cossacks" shot down #MH17 URL **[18]:** @name1 "indonesian student" ??? my poor fellow countrymen ... #damn you terrorist! **(Cluster 2) [4]:** @name1 such violence and hate but we will be celebrating Mandela's Birthday tomorrow with acts of giving and love in South Africa #peace **[17]:** @name1 Another Hollywood story is in making :-) CIA should make a story bank. Poor Hollywood is struggling for JamesBond films. @name5 **[20]:** @name1 Jim, Jim, Jim, shame on you. It was a hoax of sorts and you promoted it Opps there goes our credibility again Too bad @name6 **(Cluster 3) [1]:** @name1 this is beyond insane... **[2]:** @name1 Leaked by whom? Or don't you bother verifying sources? **[3]:** 親露派指導者と露諜報機関の通話「コサックが撃墜した」 RT @name1 NEW. Leaked phone call between rebel leader & Russian ... URL URL **[9]:** @name1 **[10]:** @name1 checked that? **[11]:** @name1 @name7 @name8 **[13]:** @name1 @name9 Thus it begins... **[16]:** @name10 @name11 @name1 ありゃりゃ 、 もう言い逃れできないわな 。 **[19]:** @name1 @name5 what is the proof |
| | **Extractive Summary [1]:** NEW. Leaked phone call between rebel leader & Russian intel agent: " Cossacks" shot down #MH17 URL **[2]:** @name1 Jim, Jim, Jim, shame on you. It was a hoax of sorts and you promoted it Opps there goes our credibility again Too bad @name6 **[3]:** @name1 @name9 Thus it begins... |
| | **Abstractive Summary [1]:** RT: NEW. Leaked phone call between rebel leader & Russian intel agent: "Cossacks" shot down #MH17 **[2]:** This is a hoax, it was a hoax of sorts and you promoted it. Shame on you **[3]:** what is the proof?? |
| 16 | **Source Post (True Rumor):** microsoft is reportedly buying 'minecraft' developer mojang for $2 billion URL |
| | **Responses (Cluster 1) [2]:** @name1 Good job, @name2 **[8]:** @name1 @name3 NOOO **[12]:** @name1 that's one way to get it onto Windows Phone. **[15]:** @name1 Pls no. **[14]:** @name1 NOOO!!! **[18]:** @name1 don't do it mojang!!! ... **[19]:** @name1 oh god why.. **(Cluster 2) [1]:** @name1 For some reason I don't believe that. They turned down offers before, so i'm sure they will again. **[3]:** . "@name1: Microsoft is reportedly buying 'Minecraft' developer Mojang for $2 billion URL" **[5]:** "@name1: Microsoft is reportedly buying 'Minecraft' developer Mojang for $2 billion URL" @name4 **[6]:** "@name1: Microsoft is reportedly buying 'Minecraft' developer Mojang for $2 billion URL" wauw! @name5 **[7]:** "@name1: Microsoft is reportedly buying 'Minecraft' developer Mojang for $2 billion URL? **[10]:** "@name1: Microsoft is reportedly buying 'Minecraft' developer Mojang for $2 billion URL" whoa **[13]:** "@name1: Microsoft is reportedly buying 'Minecraft' developer Mojang for $2 billion URL" Wat? **[20]:** @name6 "@name1: Microsoft is reportedly buying 'Minecraft' developer Mojang for $2 billion URL" **(Cluster 3) [4]:** @name7 @name8 @name1 better buy minecraft now before it's off the psn store **[9]:** @name9 wooottt? Have you ever played Minecraft? That game will eat up your time I swear and babu objectives just dig up sand and create **[11]:** @name7 @name8 @name1 lol already on there. **[16]:** @name10 @name1 @name3 @name2 From what I'm told, Notch walked away from Mojang. He's doing things on his own now. **[17]:** @name11 @name7 @name8 @name1 yeah microsoft |
| | **Extractive Summary [1]:** @name1 Pls no. **[2]:** "@name1: Microsoft is reportedly buying 'Minecraft' developer Mojang for $2 billion URL" whoa **[3]:** @name7 @name8 @name1 lol already on there. |
| | **Abstractive Summary [1]:** NOOO!!!... **[2]:** Microsoft is reportedly buying 'Minecraft' developer Mojang for $2 billion **[3]:** I think Minecraft is a great game. |

Table 11: Generated examples of DAS ($k$=3). QID corresponds to the Question ID in Fig. 6. The responses are arranged in different clusters and chronological order. Key information captured by summaries is highlighted with different colors for each cluster. The responses within the same cluster deliver similar information, and the produced summaries can effectively capture essential information from the responses.