# OpenReview forum: "Beyond Detection: A Defend-and-Summarize Strategy for Robust and Interpretable Rumor Analysis on Social Media"
_EMNLP/2023/Conference — EMNLP 2023 Main_

### Official Review · Reviewer_v8Le · 2023-08-03

**Soundness:** 4

**Excitement:**

3: Ambivalent: It has merits (e.g., it reports state-of-the-art results, the idea is nice), but there are key weaknesses (e.g., it describes incremental work), and it can significantly benefit from another round of revision. However, I won't object to accepting it if my co-reviewers champion it.

**Paper Topic And Main Contributions:**

This paper proposes a defend-and-summarize strategy for robust and interpretable fake news detection. Specifically, it leverages abstractive and extractive summarization as interpretability approaches and employs bidirectional graph neural networks for text encoding and classification. Experiments on three datasets demonstrate that the proposed approach outperforms various existing graph-based detection approaches, could alleviate the impact of adversarial responses, and offers interpretation through generated summaries.

**Reasons To Accept:**

+ fake news detection is an important research topic
+ using summarization models in this work is novel and interesting
+ an adversarial view to fake news detection is valuable

**Reasons To Reject:**

- I wonder if it might be possible to look at more recent fake news datasets at *ACL conferences beyond 2019, as the landscape of misinformation could change drastically through time.

- The baseline approaches in this work seem to be solely graph-based approaches analyzing the texts and social media structure. I wonder if non-graph approaches might also be considered as baselines for a fair comparison.

- Section 4.3's title reads "interpret predictions with summary", while the section conducts automatic and human evaluation of the quality of generated summaries. I wonder if the authors could better connect summary with interpretability, specifically why the summaries of social media posts provide interpretable evidence of model decision making. Better connecting and highlighting this in the writing is key to motivating the proposed approach.

- Which evaluation metric is adopted in Figure 3? I suggest highlighting that in the caption.

- I suggest providing more information about human evaluation in the appendix. How many annotators are employed? What's the agreement metric among them? What's the compensation scheme? etc.

- I suggest having averaged results over multiple random seeds, variation, and statistical significance tests for Table 4 since the performance gain is sometimes marginal and GNNs could be sensitive to parameter initialization.

- I wonder if it is within ethical research practices to present specific social media posts in the paper with user ids not anonymized? There might be concerns about GDPR rights, etc.

**Reproducibility:**

4: Could mostly reproduce the results, but there may be some variation because of sample variance or minor variations in their interpretation of the protocol or method.

**Reviewer Confidence:**

4: Quite sure. I tried to check the important points carefully. It's unlikely, though conceivable, that I missed something that should affect my ratings.

---

> ### Author Rebuttal · Authors · 2023-08-29
>
> Dear Reviewer v8Le,
> We really appreciate the detailed and insightful comments. We have followed the suggestions to revise the paper accordingly. Furthermore, we would like to clarify several points as follows.
>
> 1. **More recent fake news datasets at \*ACL conferences beyond 2019** \
> Firstly, we would like to point out that our work is centered around the task of rumor detection rather than fake news detection. The distinction between these two tasks lies in their data sources. Rumor detection primarily draws data from social media platforms, while fake news detection focuses more on analyzing formal news content. Moreover, the intricate nature of social media adds a layer of complexity to data formatting, including structural relationships among responses and the data collection process itself. To the best of our knowledge, BEARD [6] is the latest and only rumor dataset released at *ACL conferences after 2019. However, this dataset focuses on early rumor detection and does not contain structural information for each conversation thread. Furthermore, recent studies [3, 4, 5] highlight that RE2019, Twitter15, and Twitter16 are the most frequently used datasets for rumor detection. We therefore follow the literature to benchmark our methods on these datasets.
> 2. **Consider non-graph baselines for rumor detection.** \
> After several works demonstrate outstanding rumor detection performance by incorporating the graph structure of conversation threads [1, 2], recent studies typically design graph-based methods to strive for better results. For example, [2] showcases that GCN serves an important role in rumor detection compared to text-based detectors such as RNN-based models. Moreover, we would like to clarify that we have also removed the GCN component in the ablation studies (section 4.4 and Table 4). This ablation can also serve as a non-graph transformer-based baseline similar to the design of [10]. The results manifest the necessity of utilizing GCN in the rumor detection task, as removing GCN from both directions yields an average performance drop of approximately 0.7%.
> 3. **In section 4.3 “Interpret Predictions with Summary”, better connecting summary with interpretability in the writing is key to motivating the proposed approach.** \
> Thank you for the insightful suggestion. Our primary objective in the discussion of human evaluation part B is to clarify the interpretability of the summaries as human decisions. The results illustrate that decisions made based on both the responses and summaries yield an approximate accuracy and a Pearson correlation of 0.54. This characteristic is further highlighted through case studies, where the summaries effectively capture evidential information from the responses. This aids readers in substantiating the authenticity of the source post. Concerning the interpretability of model decision-making, we can refer to Table 8 in the appendix. This table demonstrates a comparable level of accuracy when using summaries compared to using responses as inputs for the detector, demonstrating similar properties as in the human evaluation. We highly agree that the relationship between summaries and interpretability serves as an important aspect of future research. We have taken your suggestions into account and revised the descriptions in accordance with the updated version.
> 4. **Highlight the evaluation metric of Fig. 3 in the caption.** \
> The evaluation metrics used in Figure 3 are macro-$F_1$ and attack success rate (ASR) for the left and right figures respectively. Specifically, the metrics are marked on the top for both figures. We will add the metrics to the caption for further clarification as follows. \
> “*__Caption of Figure 3__: Effect of extract ratio $\rho$ and number of clusters $k$ on Twitter15. The dashed lines represent the detection performance without DAS. The Macro-F1 (left) increases as the extract ratio $\rho$ decreases on both datasets and the Attack Success Rate (right) behaves in an opposite trend. Moreover, the number of clusters $k$ does not influence the results significantly, demonstrating the robustness of DAS.*”
> 5. **Provide more information about human evaluation in the Appendix.** \
> We have provided the details of human evaluation in the appendix according to your suggestions as follows. \
> “*We recruit 100 human readers for both parts of the human evaluation. We additionally insert a trick question in part A to validate the answer quality of each participant. Each participant will be paid \$5 if he/she passes the trick question. To verify the consistency of the collected results, we calculate the Fleiss' Kappa to evaluate the inter-rater reliability of the human evaluation. For part A, we calculate the Fleiss' Kappa to assess the readers' agreement toward rating our model more favorably than the baseline, and the score is 0.3321. In part B, for readers that make predictions based on the responses and summary, we obtain the Fleiss' Kappa scores of 0.5341 and 0.4225, respectively. Following the criteria outlined by [7], these values indicate a fair and moderate agreement among participants in part A and B. It's important to note that this level of agreement has been established as reliable in prior research studies [8, 9].*”
> 6. **Provide statistical significance tests for Table 4.** \
> All of our experiments are conducted with 5-fold cross-validation including the results in Table 4. We have provided the results of significance tests in the revised version as follows. \
> “*To validate the significance of the detection results, we perform the paired student t-test between the proposed BiTGN and each of the baseline models. The corresponding $p$-values for both accuracy and macro-averaged $F_1$ scores are presented in the following table.*
>
>     |Model|RE2019||Twitter15||Twitter16||
>     |-----|---|---|-----|----|-----|----|
>     |     |$p$ (Acc.)|$p$ (m$F_1$)|$p$ (Acc.)|$p$ (m$F_1$)|$p$ (Acc.)|$p$ (m$F_1$)|
>     |RvNN |$1.29\times10^{-4}$|$1.30\times10^{-4}$|$6.79\times10^{-5}$|$7.60\times10^{-5}$|$1.14\times10^{-4}$|$1.15\times10^{-4}$|
>     |BiGCN|$9.29\times10^{-3}$|$5.52\times10^{-3}$|$3.14\times10^{-3}$|$1.42\times10^{-3}$|$1.31\times10^{-1}$|$1.30\times10^{-1}$|
>     |EBGCN|$4.14\times10^{-3}$|$4.25\times10^{-3}$|$1.29\times10^{-3}$|$1.38\times10^{-3}$|$2.52\times10^{-2}$|$2.39\times10^{-2}$|
>     |WETGN|$2.25\times10^{-1}$|$1.69\times10^{-1}$|$3.32\times10^{-1}$|$3.37\times10^{-1}$|$1.27\times10^{-1}$|$1.37\times10^{-1}$|
>     |DUCK$_{\lnot\text{UT}}$|$3.57\times10^{-1}$|$3.57\times10^{-1}$|$1.13\times10^{-1}$|$1.00\times10^{-1}$|$1.09\times10^{-1}$|$1.12\times10^{-1}$|
>
>     *Notably, our BiTGN not only achieves the best average accuracy and $F_1$ scores as discussed previously but also significantly outperforms models utilizing TF-IDF vectors as node features (BiGCN, EBGCN, RvNN). We notice that the significance levels for the comparisons with WETGN and DUCK$\_{\lnot\text{UT}}$ are relatively lower. This could be attributed to their shared utilization of a transformer backbone and the notable variability in folds. Nonetheless, in comparison to WETGN, which incorporates a top-down GCN with weighted edges, our BiTGN effectively benefits from the bidirectional GCN component, even in the absence of weighted edges. When compared against DUCK$_{\lnot\text{UT}}$, which employs two distinct branches of transformers to independently model each conversation thread as both a stream and a graph, our model achieves a more favorable average performance by utilizing only a single transformer branch, thereby resulting in fewer parameters.*”
> 7. **Should anonymize the user IDs in social media posts presented in the paper.** \
> Thank you for bringing this to our attention. We recognize the utmost significance of safeguarding users' privacy. To begin with, it's worth noting that all user IDs related to both source posts and responses in the social media content are anonymized in the original paper. Concerning the IDs mentioned within the response content (@'s), we have taken steps to anonymize them as well. The revision will be in the following format: @PolitiFact @joerogan → @name1 @name2.
>
> [1] Jing Ma, Wei Gao, and Kam-Fai Wong. 2018. Rumor detection on Twitter with tree-structured recursive neural networks. In Proceedings of the 56th Annual Meeting of the Association for Computational Linguistics (Volume 1: Long Papers), pages 1980–1989, Melbourne, Australia. Association for Computational Linguistics. \
> [2] Tian Bian, Xi Xiao, Tingyang Xu, Peilin Zhao, Wen-bing Huang, Yu Rong, and Junzhou Huang. 2020. Rumor detection on social media with bi-directional graph convolutional networks. In Proceedings of the AAAI conference on artificial intelligence, volume 34, pages 549–556. \
> [3] Lin Tian, Xiuzhen Zhang, and Jey Han Lau. 2022. DUCK: Rumour detection on social media by modelling user and comment propagation networks. In Proceedings of the 2022 Conference of the North American Chapter of the Association for Computational Linguistics: Human Language Technologies, pages 4939–4949, Seattle, United States. Association for Computational Linguistics. \
> [4] Tiening Sun, Zhong Qian, Sujun Dong, Peifeng Li, and Qiaoming Zhu. 2022. Rumor detection on social media with graph adversarial contrastive learning. In Proceedings of the ACM Web Conference 2022, pages 2789–2797. \
> [5] Ruichao Yang, Jing Ma, Hongzhan Lin, and Wei Gao. 2022. A Weakly Supervised Propagation Model for Rumor Verification and Stance Detection with Multiple Instance Learning. In Proceedings of the 45th International ACM SIGIR Conference on Research and Development in Information Retrieval (SIGIR '22). Association for Computing Machinery, New York, NY, USA, 1761–1772. \
> [6] Fengzhu Zeng and Wei Gao. 2022. Early Rumor Detection Using Neural Hawkes Process with a New Benchmark Dataset. In Proceedings of the 2022 Conference of the North American Chapter of the Association for Computational Linguistics: Human Language Technologies, pages 4105–4117, Seattle, United States. Association for Computational Linguistics. \
> [7] Landis, J. Richard, and Gary G. Koch. “The Measurement of Observer Agreement for Categorical Data.” Biometrics 33, no. 1 (1977): 159–74. \
> [8] Shuyang Cao and Lu Wang. 2021. CLIFF: Contrastive Learning for Improving Faithfulness and Factuality in Abstractive Summarization. In Proceedings of the 2021 Conference on Empirical Methods in Natural Language Processing, pages 6633–6649, Online and Punta Cana, Dominican Republic. Association for Computational Linguistics. \
> [9] Xiuying Chen, Hind Alamro, Mingzhe Li, Shen Gao, Xiangliang Zhang, Dongyan Zhao, and Rui Yan. 2021. Capturing Relations between Scientific Papers: An Abstractive Model for Related Work Section Generation. In Proceedings of the 59th Annual Meeting of the Association for Computational Linguistics and the 11th International Joint Conference on Natural Language Processing (Volume 1: Long Papers), pages 6068–6077, Online. Association for Computational Linguistics. \
> [10] Khoo, L. M. S., Chieu, H. L., Qian, Z., & Jiang, J. (2020). Interpretable Rumor Detection in Microblogs by Attending to User Interactions. Proceedings of the AAAI Conference on Artificial Intelligence, 34(05), 8783-8790.

---

### Official Review · Reviewer_PMQb · 2023-08-04

**Soundness:** 4

**Excitement:**

3: Ambivalent: It has merits (e.g., it reports state-of-the-art results, the idea is nice), but there are key weaknesses (e.g., it describes incremental work), and it can significantly benefit from another round of revision. However, I won't object to accepting it if my co-reviewers champion it.

**Paper Topic And Main Contributions:**

This paper aims to solve two challenges: (1) improve the robustness of rumor detection models against critical responses (2) improve the interpretability of deep rumor detection models. To solve these two challenges, the authors propose a rumor detection model called Defend-And-Summarize (DAS) that is robust against malicious responses. Additionally, DAS can provide explanations to its prediction results with extractive and abstractive summaries. Their experiment results on three public datasets show that DAS is resilient against attacks and can generate multi-perspective explanations.

**Reasons To Accept:**

•	This paper clearly identifies two key drawbacks of existing rumor detectors, lack of robustness against critical responses and lack of interpretability.
•	The experiment results are complete and can support the claims. Overall, the experiments seem to be reproducible. The proposed method outperforms existing methods.
•	Overall, this paper tries to solve important issues associated with rumor detection, and provides insights for future work.


**Reasons To Reject:**

•	The response extractor and abstractor is built upon the assumption that “responses with similar stances or viewpoints should lie closer in the embedding space.” However, this assumption is made without any previous literature support or experiment support.
•	Certain claims are made without justification. For example, in section 3.2, the authors claim transformer encoders are robust without any literature or data support.
•	The captions for the figures are not informative. Therefore, the figures are hard to comprehend.


**Reproducibility:**

4: Could mostly reproduce the results, but there may be some variation because of sample variance or minor variations in their interpretation of the protocol or method.

**Reviewer Confidence:**

3: Pretty sure, but there's a chance I missed something. Although I have a good feel for this area in general, I did not carefully check the paper's details, e.g., the math, experimental design, or novelty.

**Typos Grammar Style And Presentation Improvements:**

The authors mentioned “to improve detector vulnerability” at multiple places (e.g. line 92). I think the authors meant to say “reduce detector vulnerability” or “improve detector robustness”.

---

> ### Author Rebuttal · Authors · 2023-08-29
>
> Dear Reviewer PMQb, \
> We appreciate the comments and suggestions provided in your review. We have revised the mentioned parts in our paper and would like to clarify several points in the following.
>
> 1. **The assumption of extractor and abstractor (“responses with similar stances or viewpoints should lie closer in the embedding space”) is made without previous literature support or experiment support.** \
> Thank you for highlighting this matter. We have provided the following descriptions and citations to support this assumption in the revised version. \
> “*The design of DAS follows the idea that responses with similar stances or viewpoints should lie closer in the embedding space. This concept is substantiated by prior studies [3, 4], which showcase that various standpoints of political opinions on Twitter can be well partitioned into distinct clusters based on the embedding representations. This characteristic could enhance summarization with more structured and comprehensive information.*”
> 2. **Certain claims are made without justification (section 3.2 “transformer encoders are robust” is claimed without any literature or data support).** \
> Thanks for the comment. In the original paper, we aim to emphasize the importance of robust textual representations. We have added the following descriptions and citations in section 3.2 to enhance the claim in the revised version. \
> “*Previous studies have shown that Transformer-based models are more robust to out-of-distribution data [1] and adversarial attacks [2] compared to conventional models such as CNN and RNNs.*”
> 3. **The captions for the figures are not informative.** \
> Thank you for pointing this out. We have provided more detailed information for each figure in the revised version as follows. \
> “*__Caption of Figure 1__: Three examples for the predicted probability of each class with respect to the responses on the Twitter15 dataset. Curves with their face colored represent the ground-truth labels for their source post. Critical responses that result in prediction shifts larger than 0.5 are marked with a red circle. \
> __Caption of Figure 2__: Overview of our proposed framework (upper left). The rumor detector BiTGN (upper right) is trained to predict the veracity of each source post. The response summarizer DAS (lower) preemptively filters out attack responses generated by the response generator. It then organizes the remaining responses into $k$ clusters and produces both extractive and abstractive summaries for each cluster accordingly. During inference, the detector makes predictions based on the source post and the summaries. \
> __Caption of Figure 3__: Effect of extract ratio $\rho$ and number of clusters $k$ on Twitter15. The dashed lines represent the detection performance without DAS. The Macro-F1 (left) increases as the extract ratio $\rho$ decreases on both datasets and the Attack Success Rate (right) behaves in an opposite trend. Moreover, the number of clusters $k$ does not influence the results significantly, demonstrating the robustness of DAS. \
> __Caption of Figure 4__: Human evaluation of generated summaries. In part A, our SSRA-$k$-means model can generate more informative response summaries compared to SSRA-LOO. In part B, human predictions based on either responses or summaries can achieve comparable accuracy, which demonstrates the interpretability of the summaries.*”
> 4. **“Improve detector vulnerability” $\rightarrow$ “reduce detector vulnerability” or “improve detector robustness”** \
> Thanks for the correction. We have modified the descriptions accordingly in the revised version.
>
>
> [1] Dan Hendrycks, Xiaoyuan Liu, Eric Wallace, Adam Dziedzic, Rishabh Krishnan, and Dawn Song. 2020. Pretrained Transformers Improve Out-of-Distribution Robustness. In Proceedings of the 58th Annual Meeting of the Association for Computational Linguistics, pages 2744–2751, Online. Association for Computational Linguistics. \
> [2] Jin, Di, et al. "Is bert really robust? a strong baseline for natural language attack on text classification and entailment." Proceedings of the AAAI conference on artificial intelligence. Vol. 34. No. 05. 2020. \
> [3] Darwish, Kareem. "Quantifying polarization on twitter: the kavanaugh nomination." Social Informatics: 11th International Conference, SocInfo 2019, Doha, Qatar, November 18–21, 2019, Proceedings 11. Springer International Publishing, 2019. \
> [4] Rashed, A., Kutlu, M., Darwish, K., Elsayed, T., & Bayrak, C. (2021). Embeddings-Based Clustering for Target Specific Stances: The Case of a Polarized Turkey. Proceedings of the International AAAI Conference on Web and Social Media, 15(1), 537-548.

---

### Official Review · Reviewer_4y82 · 2023-08-05

**Soundness:** 4

**Excitement:**

3: Ambivalent: It has merits (e.g., it reports state-of-the-art results, the idea is nice), but there are key weaknesses (e.g., it describes incremental work), and it can significantly benefit from another round of revision. However, I won't object to accepting it if my co-reviewers champion it.

**Missing References:**

None come to mind.

**Paper Topic And Main Contributions:**

The paper proposes a novel framework in the interpretation of rumors and conspiracy theories on social media caled Defend and Summarize or DAS. The key principle here is that by focusing the attention on responses and adding summarization modules of varying levels of abstraction, we can get multiple perspectives to be able to detect rumors and also understand why they were rumors in the first place.

The model architecture contains a Transformer backbone, along with a Graph Convolutional Head and the performance of DAS is demonstrated across popular rumor corpuses that have been previously released.

**Questions For The Authors:**

A: It is usually the latent space representations that are clustered. Why is it that the output representations in DAS encoder-decoder are clustered?

B: Can we find a way to simply or clarify Fig. 2?




**Reasons To Accept:**

A: Few papers in this track attempt to formalize the task and write out the entire methods in a clear and concise mathematical framework.

B: I really appreciate the ablation study in Table 4 claiming that the GCN complex is necessary to boost performance (although very little compared to the second strongest baseline).

C: I find the pseudo-summary setup in the training of SSRA-LOO to be inspired and useful and I look to use that in my own future work.


**Reasons To Reject:**

A: The complexity of the system is oversold which affects the readability of the paper. Consider Figure 2: $x_1, \dots, x_5$ can be simply written as $x$, and a small block can imply encoding $x \rightarrow h$. DAS finds two summaries: "extractive" - from the text, "abstractive" - summarizes in own words. This is facilitated by BART.

In simple terms, the rumor detector uses summaries from the text to predict whether it is a rumor or not. The novelty appears to be in the method in which these summaries are constructed; Argument being that the points corresponding to sentences closer to each other in a latent space are more useful to be summarizes jointly.

B: In the first read, it wasn't clear to me why this method is only applied or applicable to response summarizes. Can't this framework be extended to most domains?

**Reproducibility:**

3: Could reproduce the results with some difficulty. The settings of parameters are underspecified or subjectively determined; the training/evaluation data are not widely available.

**Reviewer Confidence:**

4: Quite sure. I tried to check the important points carefully. It's unlikely, though conceivable, that I missed something that should affect my ratings.

**Typos Grammar Style And Presentation Improvements:**

I might have missed a few spots. Please find the suggestions below:
A: Examples up front will improve the readability of the paper which otherwise comes across as dense.
B: Lines 284 - 291 need clarification in the Methods.

---

> ### Author Rebuttal · Authors · 2023-08-29
>
> Dear Reviewer 4y82, \
> We sincerely appreciate your comments and suggestions. We have revised the mentioned parts in our paper and would like to clarify several points as follows.
>
> 1. **The complexity of the system is oversold which affects the readability of the paper.** \
> Thank you for your suggestions. Figure 2 may convey an excessive level of detail. Accordingly, we have made necessary modifications to the figure as follows: Firstly, for both the overall framework (upper left) and the rumor detector architecture (upper right), we have simplified the notation of input from $x_1, …, x_5$ to $x$. Additionally, we have added two blocks to represent the input and output embedding layers. For the DAS framework (lower), we preserve $h_1, …, h_6$ to illustrate the clustering mechanism performed on multiple responses. To enhance the readability, we have also included a dashed arrow connecting $h_1, .., h_6$ in the lower figure (DAS) to emphasize that the encoder-decoder within the extractor is solely employed to filter the responses. The revised figure can be found here: https://anonymous.4open.science/r/EMNLP2023-2B9A/model_overview.png
> 2. **Sentences closer to each other in a latent space are more useful to be summarized jointly.** \
> Here, we want to clarify the motivation for summarizing sentences closer in a latent space. To be more precise, our corpus does not include any summary labels, and the challenge in self-supervised summarization is how to construct an appropriate pseudo-summary. In this paper, instead of randomly selecting one response from all responses as the pseudo summary repeatedly, we propose to choose responses from clusters as the self-supervised learning signal. In a social media conversation thread, the diverse utterances of different users increase the difficulty of summarizing a specific aspect from any other perspective. Consider an extreme scenario where a conversation thread includes mostly supportive opinions and a single denying response toward the source post. If the denying one is selected as the pseudo-summary, it is unreasonable and injects substantial noise and difficulty in training a summarizer. Therefore, leveraging a clustering mechanism to aggregate similar utterances becomes pivotal, as this strategy could enhance the reliability of the pseudo-summary construction for providing diverse viewpoints within a conversation thread.
> 3. **Not clear why this method is only applied or applicable to response summarization. Can’t this framework be extended to most domains?** \
> We choose to develop this method for the rumor detection task for two main reasons. Firstly, given the prevalent use of social media nowadays, creating malicious responses to mislead model predictions becomes remarkably effortless, which leads to severe security concerns for detection models. Secondly, considering that readers can be easily influenced by one-sided information, enhancing the interpretability of detection models can not only aid readers in comprehending model behavior but also encourage the public to thoroughly consider various viewpoints before validating unverified rumors. Our proposed method aims to improve the detector's robustness by filtering out malicious responses and provide interpretability by summarizing different opinions in a conversation thread. As such, these methods can be generalized to other tasks that rely on various user feedback to make the prediction. For instance, in the context of review-based recommendation systems, we could employ the concept of response summarization to enhance attack resistance against malicious reviews by filtering and summarizing the user reviews of a given product. We will include the above discussion in our revision to shed light on future research.
> 4. **Why do we cluster the output representations of DAS?** \
> It is worth noting that we employ the original token embeddings for clustering instead of the latent representations from intermediate layers or the output representations generated by the encoder-decoder within the DAS. The encoder-decoder framework is adopted to detect the anomalies based on the reconstruction error. After filtering out the anomalous responses, we then use the original token embedding for response clustering. While it is applicable to use the latent representation from the encoder-decoder model for clustering, we found that using token embedding is sufficient to yield successful results in consolidating responses with similar content. Therefore, we choose to use the token embedding to keep our approach succinct. Furthermore, we have modified Figure 2 to better illustrate our approach.
> 5. **Can we find a way to simplify or clarify Figure 2?** \
> Thank you for your advice. We have made modifications to Figure 2 as mentioned above to enhance its simplicity and clarity.
> 6. **Examples up font will improve the readability of the paper which otherwise comes across as dense.** \
> Thanks for pointing this out, we will try to scale up the font size of the examples in the new version of our paper.
> 7. **Lines 284-291 (ARG) need clarification in the Methods.** \
> Thanks for the comment. We did not include the details of ARG in the submitted version due to the space limitations. In the revised version, we will provide more details in the appendix as follows: \
> "*To further simulate the attack responses produced by different users in real-world scenarios, we adopt an adversarial response generator (ARG) proposed by [1], which is trained by adversarial learning under white-box settings. We initialize ARG with a BART model due to its outstanding performance on several text generation tasks. Given a conversation thread $\\{x_i\\}^n_{i=0}$, the goal is to generate an adversarial response  $x_n^\ast$ that makes the detector deviate from the ground-truth $\mathbf{y}$ by maximizing the detection loss $L_{det}$, detailed in Sec. 3.5, i.e.,*
>
>     $max_{\mathbf{h_n^\ast}}(L_{det}(\mathbf{y}^\ast, \mathbf{y}))$, \
>     $\mathbf{y}^\ast=\theta_{gcn}(\theta_{enc}([\mathbf{H}^{(0)} || \mathbf{h_n^\ast}]), \mathbf{A}')$,
>
>     *where $\mathbf{h_n^\ast}$ denotes the hidden representation of $x_n^\ast$ and $\mathbf{A'}$ is the new adjacency that attaches $x_n^\ast$ to source post $x_0$. To generate a response, we construct the ARG by sharing the encoder  $\theta_{enc}$ with BiTGN and feeding the hidden representation of the last encoder layer $\mathbf{H}^{(L_{e})}$ to the decoder $\theta_{dec}$ as:*
>
>     $\mathbf{h_n^\ast}=\theta_{dec}(\mathbf{H}^{(L_e)})$, \
>     $x_n^\ast=argmax(softmax(\theta_{out}(\mathbf{h_n^\ast})))$.
>
>     *Note that $\theta_{out}$ denotes the output layer, which is tied with the input embedding layer $\theta_{in}$. In this way, $\mathbf{h_n^\ast}$ can approximate the embedding of a generated response and be concatenated with the embedding of $\\{x_i\\}^n_{i=0}$ that serves as part of the encoder's inputs without taking $argmax$ operation. Subsequently, the gradients can be backpropagated from the rumor detection loss to train the ARG. Moreover, the generated response is attached to the source post of the thread and a new edge $e_{0,n}$ between $x_0$ and $\widehat{x}_n$ is thus created.*"
>
> [1] Yun-Zhu Song, Yi-Syuan Chen, Yi-Ting Chang, Shao-Yu Weng, and Hong-Han Shuai. 2021. Adversary-Aware Rumor Detection. In Findings of the Association for Computational Linguistics: ACL-IJCNLP 2021, pages 1371–1382, Online. Association for Computational Linguistics.

---

### Meta-Review · Area_Chair_jqXr · 2023-09-15

**Recommendation:** 4

**Metareview:**

A framework for interpretation of rumors and cospiracy theory has been proposed. Its main principles are focused on the attention on response and to analyze summarization module.

This paper clearly identifies two key drawbacks of existing rumor detectors, lack of robustness against critical responses and lack of interpretability.  Furthermore, the task has been well defined and the experiment results are complete, also providing an interesting ablation study

Some concerns are related to the evaluation of non-graph approaches might also be considered as baselines for a fair comparison, also providing more information about human evaluation in the appendix

---

### Decision · Program_Chairs · 2023-10-07

**Decision:**

Accept-Main

**Comment:**

A framework for interpretation of rumors and cospiracy theory has been proposed. Its main principles are focused on the attention on response and to analyze summarization module.

This paper clearly identifies two key drawbacks of existing rumor detectors, lack of robustness against critical responses and lack of interpretability.  Furthermore, the task has been well defined and the experiment results are complete, also providing an interesting ablation study

Some concerns are related to the evaluation of non-graph approaches might also be considered as baselines for a fair comparison, also providing more information about human evaluation in the appendix